# Subtle changes in chromatin loop contact propensity are associated with differential gene regulation and expression

William W. Greenwald[1], He Li[2,8], Paola Benaglio[3], David Jakubosky[4,5], Hiroko Matsui[2], Anthony Schmitt[6], Siddarth Selvaraj[6], Matteo D'Antonio [2,7], Agnieszka D'Antonio-Chronowska[2], Erin N. Smith[3] & Kelly A. Frazer[2,3]

While genetic variation at chromatin loops is relevant for human disease, the relationships between contact propensity (the probability that loci at loops physically interact), genetics, and gene regulation are unclear. We quantitatively interrogate these relationships by comparing Hi-C and molecular phenotype data across cell types and haplotypes. While chromatin loops consistently form across different cell types, they have subtle quantitative differences in contact frequency that are associated with larger changes in gene expression and H3K27ac. For the vast majority of loci with quantitative differences in contact frequency across haplotypes, the changes in magnitude are smaller than those across cell types; however, the proportional relationships between contact propensity, gene expression, and H3K27ac are consistent. These findings suggest that subtle changes in contact propensity have a biologically meaningful role in gene regulation and could be a mechanism by which regulatory genetic variants in loop anchors mediate effects on expression.

[1] Bioinformatics and Systems Biology Graduate Program, University of California, San Diego, La Jolla, CA 92093, USA. [2] Institute for Genomic Medicine, University of California, San Diego, La Jolla, CA 92093, USA. [3] Department of Pediatrics and Rady Children's Hospital, University of California, San Diego, La Jolla, CA 92093, USA. [4] Biomedical Sciences Graduate Program, University of California, San Diego, La Jolla, CA 92093, USA. [5] Department of Biomedical Sciences, University of California, San Diego, La Jolla, CA 92093, USA. [6] Arima Genomics, San Diego, CA 92121, USA. [7] Moores Cancer Center, University of California, San Diego, La Jolla, CA 92093, USA. [8] Present address: Human Genome Sequencing Center, Baylor College of Medicine, Houston, TX 77030, USA. These authors contributed equally: William W. Greenwald, He Li. Correspondence and requests for materials should be addressed to E.N.S. (email: erinnsmith@gmail.com) or to K.A.F. (email: kafrazer@ucsd.edu)

Chromatin loops colocalize regulatory elements with their targets[1–15] by bringing genomic regions that are distant from one another in primary structure close together in 3D space[16]. These colocalized regions, also known as loop anchors, are preferentially enriched for disease associated distal regulatory variation and expression quantitative trait loci (eQTLs)[17–22]. While it has been shown that the physical 3D distance between looped loci can vary[16,23–25], previous studies examining cell type and haplotype differences in looping have considered loops to be either present or absent, rather than a quantitative phenotype. Thus, the extent to which quantitative differences between chromatin loops exist, and whether they are associated with differences in gene expression and regulation, has yet to be explored.

Bulk chromatin conformation assays (e.g., 3C, 4C, and Hi-C) were designed to measure physical contact frequency between two pieces of colocalized (i.e., looped) DNA in a pool of cells. While a recent single-cell Hi-C study found that contacts occur within single cells at loops called from bulk data, there was variability in the contact profiles of looped loci between cells[26]. Together, this suggests that the contact frequency measured in a pool of cells reflects the proportion of cells in which a contact is occurring, or the probability for the contact to occur (contact propensity) across all cells in the sample. Investigating contact frequency as measured by Hi-C, in combination with molecular phenotypes, may reveal if contact propensity between looped loci varies across cell types and haplotypes, and if this variation is associated with differential regulation of gene expression.

If contact propensity between looped loci does in fact play a role in gene regulation, a genetic variant that affects contact propensity would likely have a downstream effect on gene expression. Therefore, the association between contact propensity and gene expression would exist not only across cell types, but also across haplotypes. Recent studies examining whether chromatin loops vary across haplotypes, and the functional consequences of this variation, have come to conflicting conclusions. Rao et al.[2] created and phased the GM12878 Hi-C map (which is the highest-resolution map currently available) to study differences in looping across haplotypes, and did not observe differences between the paternal and maternal haplotypes outside of imprinted regions. Other more recent studies employing CTCF ChIA-PET[5] and H3K27ac Hi-ChIP[27] have reported that allelic imbalance in chromatin looping occurs throughout the genome. These contradictory results are likely due to the experimental design and types of effects examined in these studies. Rao et al.[2] used Hi-C data to look for large differences across haplotypes, and thus may have missed smaller effects. The studies using ChIA-PET and Hi-ChIP sought to identify allelic imbalance of all sizes, but employed experimental approaches that may be biased as they simultaneously measure either CTCF binding or regulatory region activity and chromatin looping, thereby conflating the allelic bias of the two phenotypes. A genome-wide quantitative analysis into allele-specific chromatin looping using phased Hi-C would enable the unbiased estimation of the magnitude at which contact propensity varies across haplotypes at all types of chromatin loops (rather than only those at promoters and/or enhancers). Additionally, integrating this data with phased gene expression and H3K27ac data could provide evidence that contact propensity plays a role in long-range gene expression regulation, and provide insight into how regulatory genetic variants may influence chromatin structure.

In this study, we generate a resource of phased, high resolution Hi-C chromatin maps from induced pluripotent stem cells (iPSCs) and iPSC-derived cardiomyocytes (iPSC-CMs) from seven individuals in a three-generation family for whom we have 50× whole-genome sequence (WGS), and phase gene expression (RNA-seq) and enhancer activity (H3K27ac ChIP-seq) data generated from the same iPSCs and iPSC-CMs. We identify chromatin loops, quantitatively characterize cell-type-associated looping, and find that while loops tend to be present in both cell types, some loops exhibit significantly increased contact propensity within one cell type. We show that these quantitatively-identified cell-type-associated loops (CTALs) recapitulate known biology discovered through previous qualitative comparisons of cell-type-specific loops, including being enriched for differentially expressed genes and regulatory regions, becoming more specialized throughout differentiation, and connecting distal eQTLs to their target gene. Additionally, our quantitative analyses reveal that small magnitude changes in contact propensity are proportionally associated with large changes in molecular phenotypes: an association that could not be identified by qualitative comparisons. We next examine allelic differences in contact propensity by phasing our Hi-C data, and find that haplotype-associated chromatin loops (HTALs) are highly enriched for imprinted regions or for being associated with copy-number variation, but not for eQTLs, suggesting that regulatory genetic variants do not exert large effects on chromatin contact propensity. Finally, we examine the association between differential contact propensity and differential gene expression and H3K27ac over a range of magnitudes across both cell types and haplotypes by quantitatively associating the phenotypes in aggregate across the genome. These analyses reveal a genome-wide proportional relationship between differential contact propensity and differential expression and H3K27ac that is consistent across cell types and haplotypes. Our study therefore suggests that the cellular context of a chromatin loop (i.e., cell type, genetics, etc.) affects the propensity for an interaction at a loop to occur, and that these small changes in contact propensity are associated with large functional effects. This model suggests that regulatory genetic variation could mediate its effects on gene expression through subtle modification of contact propensity at chromatin loops.

## Results

**Sample and data collection**. Molecular data was obtained from iPSCs and their derived cardiomyocytes (iPSC-CMs) from seven individuals in a three-generation family from iPSCORE (the iPSC collection for Omics REsearch)[28] (Figure 1a, Supplementary Table 1). Fibroblasts from these seven individuals were reprogrammed using non-integrative Sendai virus vectors[29], from which eleven iPSC lines were generated and subsequently differentiated into 13 iPSC-CM samples using a monolayer-based protocol[30]. From the eleven iPSC and 13 iPSC-CM samples, we generated chromatin interaction data via in situ Hi-C[2]. Additionally, from these and other iPSC and iPSC-CM samples from the same seven individuals, we integrated functional genomic data that was generated as part of a concurrent manuscript[31] (RNA-seq for gene expression, H3K27ac ChIP-seq for enhancer activity, and ATAC-seq for chromatin accessibility; Fig. 1b; see Methods) which also describes the differentiation efficiency and quality of all iPSC and iPSC-CM lines used in this study. Finally, we obtained single-nucleotide variants (SNVs) and somatic and inherited copy-number variants (CNVs) for the seven individuals from ~50× WGS and genotype arrays from previously published work[28,32].

**Identification of chromatin loops in iPSCs and iPSC-CMs**. We characterized the 3D chromatin structure of iPSCs and iPSC-CMs by identifying chromatin loops in each cell type genome-wide. From the in situ Hi-C data, we obtained 1.74 billion long-range (≥ 20 kb) intra-chromosomal contacts after aligning and filtering ~6 billion Hi-C read pairs across all 24 Hi-C samples

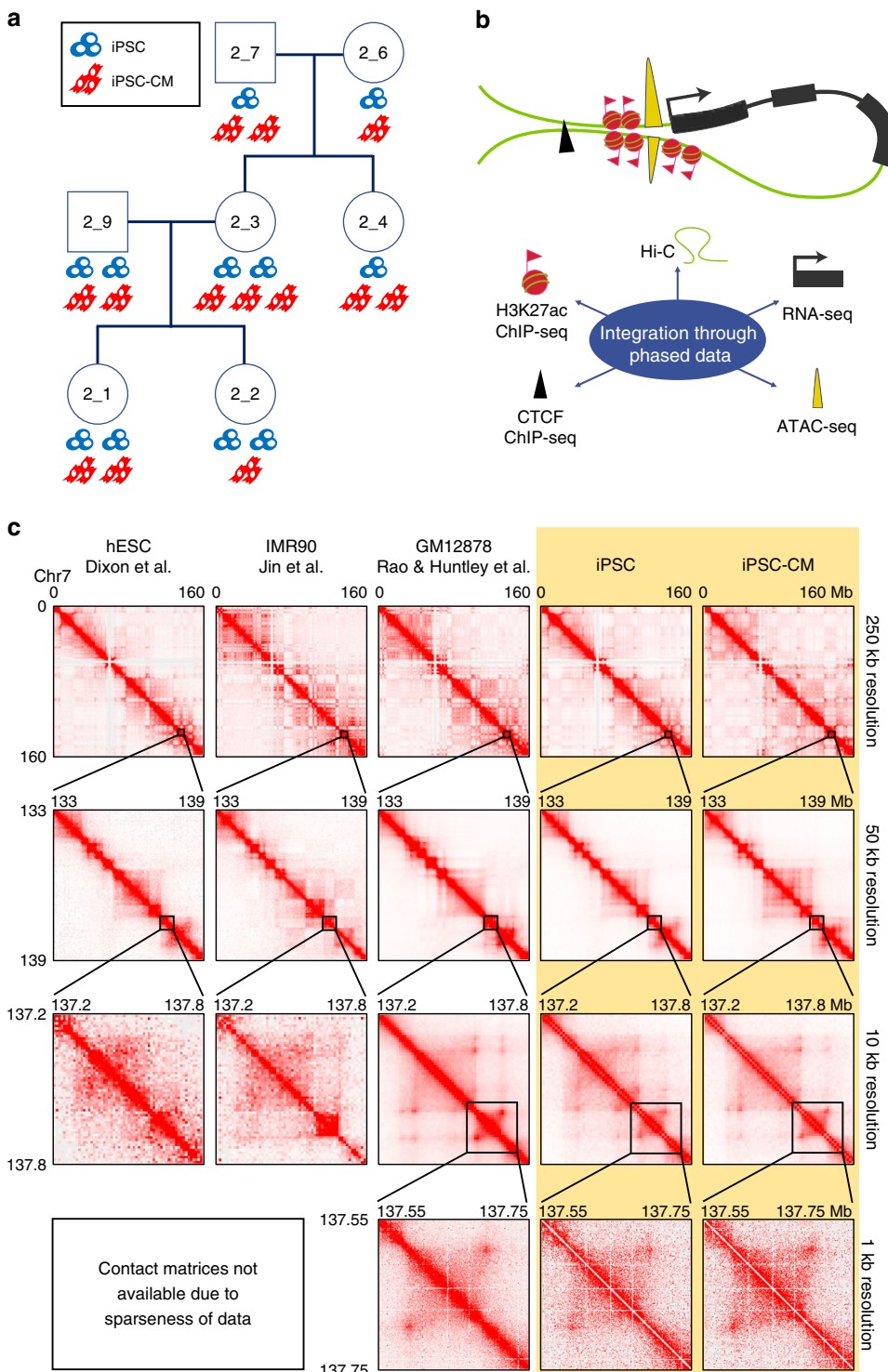

**Fig. 1** Study design, data, and chromatin contact maps. **a** Pedigree of the seven individuals used in this study. Cell icons below each subject indicate the number of iPSC lines and iPSC-CM samples used in the Hi-C experiments. iPSC lines are shown in blue, iPSC-CM samples are shown in red. **b** Schematic showing the data types used in this study depicting how they colocalize at loop anchors. **c** Hi-C contact maps from previous Hi-C studies (three left columns), and this study (two right columns, highlighted in yellow), displaying depth of map on Chromosome 7 (arbitrarily chosen for example). For Dixon et al.[10] and Jin et al.[4], the data is too sparse to zoom to 1 kb resolution

(Supplementary Figure 1A, Supplementary Table 2). We performed hierarchical clustering of the contact frequencies by cell type across individuals and observed high correlations within each cell type both by Pearson correlation, and by correcting for Hi-C biases via HiCRep[33] (Supplementary Figure 1B). To identify a set of reference loops for downstream quantitative analyses, we combined the Hi-C data within each cell type to obtain a comprehensive set of loops from high-depth data. We pooled the data

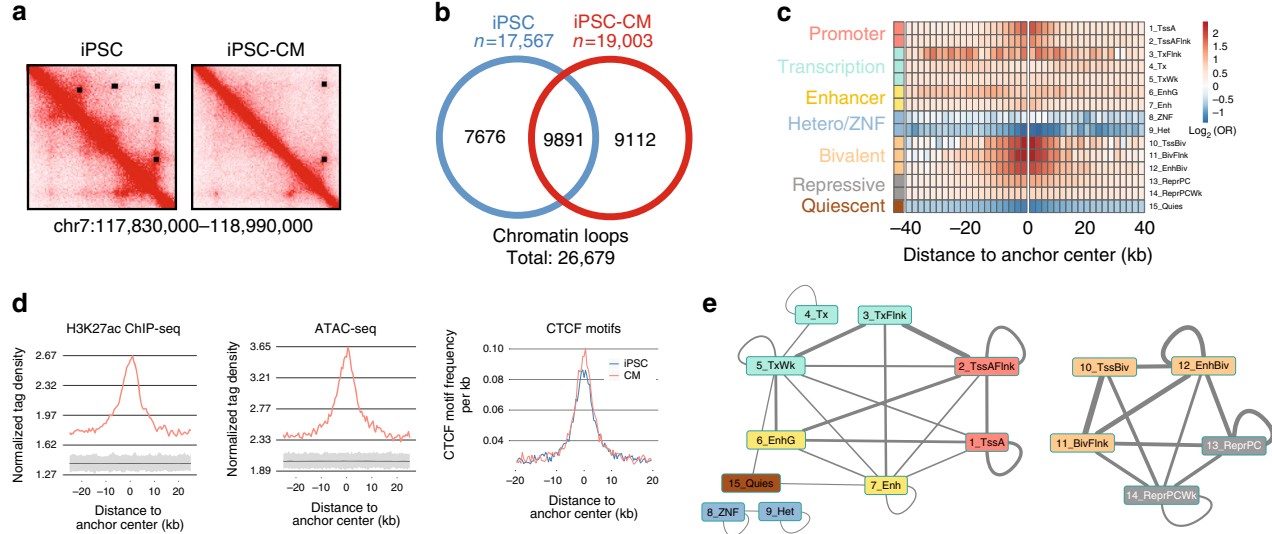

**Fig. 2** iPSC and iPSC-CM called loops. **a** Example contact maps from iPSCs (left) and iPSC-CMs (right) showing differences in looping identified by callers across cell types, with loop calls shown on the top right half of maps as black rectangles. Two loops appear present and are called in iPSCs, and four loops appear present and are called in iPSC-CMs. **b** Venn diagram showing the number of chromatin loop calls unique and common to both cell types. **c** Heatmap showing enrichment of regulatory regions near iPSC-CM called loop anchor centers. The 15 ROADMAP chromatin states of fetal-heart tissue (E083) were used, and the log$_2$ odds ratio of enrichment is indicated by color (red positive, blue negative) for each 2 kb interval across an 80 kb window. **d** Density plots showing distribution of epigenetic marks and motifs relative to the center of loop anchors. Normalized tag densities as measured by Homer from H3K27ac ChIP-seq (left) and ATAC-seq (middle) are shown for loops called in iPSC-CM. Gray regions below the peak signals indicate the results from 1000 null loop sets. CTCF motif frequency per kb (right) is shown for loops called in iPSCs (blue) or iPSC-CMs (red). **e** Network diagram showing two discrete subnetworks of fetal-heart chromatin states at iPSC-CM called loops, with edges connecting statistically significant pairs of chromatin states found at opposing loop anchors. The thickness of the edge indicates the odds ratio of enrichment, and the presence or absence of an edge indicates statistical significance

across samples for each cell type, resulting in reference chromatin maps with the highest resolution (~2 kb matrix resolution, defined as the resolution at which 80% of loci have 1000 or more contacts with any other locus[2]) in iPSCs and iPSC-CMs (or any other iPSC-derived cell type) to date, and were comparable in resolution to the Hi-C map in GM12878[2] (Fig. 1c & Supplementary Figure 1C). As loop calling algorithms often identify distinct loops, and are dependent on the resolution parameters specified for their analysis[34], we called chromatin loops from these maps utilizing two algorithms (HICCUPS and Fit-Hi-C) at multiple resolutions, identifying 17,567 loops in iPSCs (iPSC called loops), and 19,003 iPSC-CM loops (iPSC-CM called loops; Supplementary Data 1 & 2). We examined the overlap of the loop calls between cell types (Fig. 2a) and found that 37.1% of the total 26,679 loops were called in both cell types (Fig. 2b). These findings were consistent with previous studies investigating differential presence of loops between cell types[2,35]. To examine whether these loops were predominantly demarcating TADs, or were separate from TAD structure, we also called TADs in both cell types and examined the number of loops that had both anchors within 25 kb of TAD boundaries. We found only 2.9% of iPSC loops, and 5.1% of iPSC-CM loops, to have both anchors at TAD boundaries, indicating that these loops were primarily not demarcating TADs. These iPSC and iPSC-CM called loop sets provide a resource for the analysis of long-range gene regulation across the genome.

**Called chromatin loop sets contain a variety of loop types.** To characterize the types of chromatin loops that comprised the loop sets, we examined the distribution of H3K27ac and ATAC peaks, CTCF motifs, and ROADMAP chromatin states from the most epigenetically similar cell type[31] (iPSC for iPSCs; fetal heart for

iPSC-CMs) near loop anchors. In both cell types, we found enrichments for active and bivalent chromatin states (Fig. 2c & Supplementary Figure 1D), H3K27ac (Fig. 2d left & Supplementary Figure 1E), and chromatin accessibility (Fig. 2d middle & Supplementary Figure 1F) from their respective cell type above shuffled null loop sets. Additionally, we found that 45.5% of loops had CTCF motifs at both anchors, and that across all loops, CTCF motifs were centrally enriched at anchors (Fig. 2d right). As seen in Rao et al.[2], the vast majority of loops (85.3%) with CTCF motifs at both anchors had inward facing CTCF motifs. Further, 63.3% and 65.3% loops in iPSC and iPSC-CMs, respectively, were within 25 kb of a CTCF ChIA-PET interaction from GM12878[5]. We next examined the types of chromatin states that were statistically significantly paired together (Fisher's exact $p <$ 0.05) and found two subnetworks, one with active chromatin states and the other with repressed or bivalent chromatin, which were discrete in iPSC-CMs (Fig. 2e) and crossed over through the bivalent states in iPSCs (Supplementary Figure 1H). This crossover, which was only present in iPSCs, is consistent with the role of bivalent and polycomb chromatin in pluripotency[36–38], the role of bivalency in maintaining stem cell region connectivity[38], and with the shift of active states to bivalent and polycomb during differentiation and chromatin rewiring[39]. This result suggests that these specialized roles of bivalent and polycomb chromatin extend to the fine-scale aspects of chromatin architecture, including loops. We next examined the consistency of these loops with previously identified promoter loops from promoter-capture Hi-C (pHiC) and found 28.7% and 33.5% of iPSC and iPSC-CM loops to be within 25 kb of a pHiC interaction in these cell types, respectively. Together, these results indicate that the identified chromatin loops include those with active regulatory interactions (e.g., promoter–enhancer interactions), those with repressive

interactions (e.g., polycomb complexes), those that are structural (CTCF-CTCF), and those with a variety of other types of chromatin states (that were not significantly enriched for being paired together) at their anchors.

**Quantification of differential looping between cell types.** Statistical methods for finding differential loops across conditions remains a largely open question in the field of chromatin architecture[34]. We found a large proportion of loops which were differentially called, but visually appeared to consistently form across cell types (Fig. 3a). Thus, to determine if the chromatin loops called in only one of the cell types specifically formed within that cell type, or whether they were also present in the other cell type but not called for technical reasons, we performed a quantitative comparison of the subjects' contact frequencies between the iPSC and iPSC-CM using edgeR[40–42]. For all loops, identified in either one or both cell types, we first compared the total normalized contact frequency ($\log_2$ counts per million, logCPM, obtained via edgeR) of the interactions between both cell types. We observed that the majority of loops that were called in both cell types (gray in Fig. 3b left) had high logCPMs in both cell types, whereas the loops that were only called in a single cell type (blue or red in Fig. 3b left) tended to have overall low logCPMs and often showed highly similar contact intensities between cell types. We did not observe, however, loops with a high logCPM in one cell type, and a very low logCPM in the other. These patterns were similar within subjects, suggesting that these subtle modulations in logCPM across cell types were not due to the combination of data across individuals (Supplementary Figure 2). These results indicate that chromatin loops that were called as differentially present or absent between cell types were often of low logCPM, and were therefore likely to be inconsistently identified by the loop calling algorithms. Thus, the differences in the loop sets between the two cell types were not due to the establishment of novel loops present in only one cell type. We therefore identified loops that showed quantitative differences between iPSCs and iPSC-CMs by statistically comparing normalized read counts across cell types at each loop identified in either cell type (edgeR glmQLFit on trimmed mean of M values, TMMs, $q < 0.01$). These CTALs were identified across a range of logCPM levels and were distinct from those called within each cell type (Fig. 3b right). This analysis resulted in four loop sets (Supplementary Data 3): (1) all loops called in any cell type (union loop set, total: 26,679), (2) loops with statistically higher contact frequency in iPSCs (iPSC cell-type-associated loops; iPSC-CTALs, total 2906), (3) loops with statistically higher contact frequency in iPSC-CMs (CM-CTALs, total 2915), and (4) loops that were not statistically significantly different between the two cell types (non-CTALs, total 20,858). To determine whether 3D architecture at a compartment level contributed to these differences, we identified A and B compartments[2] and partitioned the loops by their location in both cell types. While we found increased contact propensity within A compartments relative to B compartments in both cell types (Supplementary Figure 3A), the percent of variance in logCPM explained by compartment differences was only 0.009 (Supplementary Figure 3B). Additionally, we found that the CTAL distribution was consistent across all types of anchor-compartment-cell type combinations (Supplementary Figure 3C). These results suggest that compartment differences did not drive CTALs. Overall, these analyses establish cell-type-associated loop sets for future analyses.

**CTALs are associated with differentiation regulatory changes.** Previous studies which qualitatively identified cell-type-specific loops have reported that chromatin architecture becomes more specialized and cell type specific during development[35,43,44]. We examined the physical and regulatory characteristics of iPSC-CTALs and CM-CTALs to determine if these quantitatively-identified loops recapitulated these same properties. We observed that CM-CTALs were overall significantly larger (Mann–Whitney $p < 2.2 \times 10^{-16}$; Fig. 3c) and more complex (i.e., shared more anchors with one another; Mann–Whitney $p < 2.2 \times 10^{-16}$; Fig. 3d) than iPSC-CTALs. Additionally, we found active chromatin states to be preferentially enriched at smaller (Fig. 3e) and less complex (Fig. 3f) loops. We examined how the enrichment of H3K27ac and ATAC-seq signals varied by CTAL status, and found that within each cell type, CTALs of that cell type and non-CTALs had the highest H3K27AC and ATAC-seq signal, while CTALs of the other cell type were least enriched (Fig. 3g). These enrichments suggest that loops with decreased contact propensity may be less likely to be involved in gene regulation despite being present in the cell. Next, we examined whether CTALs for each cell type were more likely to overlap cell type specific, or cell type shared, regulatory regions. We found iPSC-CTAL and CM-CTAL anchors to be enriched for differential active promoters, and iPSC-CTAL anchors to be enriched for differential active enhancers (Fig. 3h, red). These enrichments suggest that CTALs capture cell type specific chromatin dynamics, and are consistent with active elements shifting to repressed elements during differentiation and chromatin rewiring[39] (as enhancers from fetal heart tended to be present in both cell types, but enhancers in iPSCs tended to be iPSC specific). We also observed that iPSC-CTAL anchors which overlapped iPSC bivalent enhancers were more likely to overlap fetal heart bivalent enhancers (Fig. 3h, blue), but not the converse, consistent with the repression of active regions of loops during differentiation, and specific use of bivalent chromatin in iPSCs[36,37,39]. Overall, these findings show that CTALs were enriched for cell type specific functional and regulatory regions.

**Functional characterization of CTALs.** To analyze whether CTALs recapitulated the functional differences between qualitatively identified cell-type-specific loops, we examined the relationship between contact propensity and eQTLs, differential gene expression, and differential epigenetics across cell types. We first examined whether loops which colocalize iPSC-eQTLs (previously identified from a cohort including these individuals[32]) to the genes that they were statistically associated with (eGenes) had stronger contact intensities within iPSCs than iPSC-CMs. We found a strong enrichment (Mann–Whitney U test $p \sim 1 \times 10^{-293}$) for increased iPSC:iPSC-CM contact frequency ratio above non-eQTL-eGene loops (Fig. 4a), indicating that loops with higher contact propensity in a cell type may be more likely to harbor functional genetic variation. Next, we examined whether differential molecular phenotypes were preferentially located at CTAL anchors. We identified differential H3K27ac peaks and genes using ChIP-seq and RNA-seq data generated from iPSC and iPSC-CM samples from the same seven individuals (see Methods). We obtained a total of 23,570 differential H3K27ac peaks (DE peaks) and 5307 differential genes (DE genes) between iPSCs and iPSC-CMs. We found that DE genes and DE peaks were preferentially located at CTAL anchors (Fisher's exact $p < 0.05$, Fig. 4b) compared to the union loop set. Together, these results show that CTALs (loops with quantitative differences in contact propensity across cell types) are associated with cell-type-specific functions, and are consistent with previous reports that used qualitatively identified cell-type-specific loops.

**Subtle looping changes are associated with gene regulation.** Next, we examined the quantitative association between contact

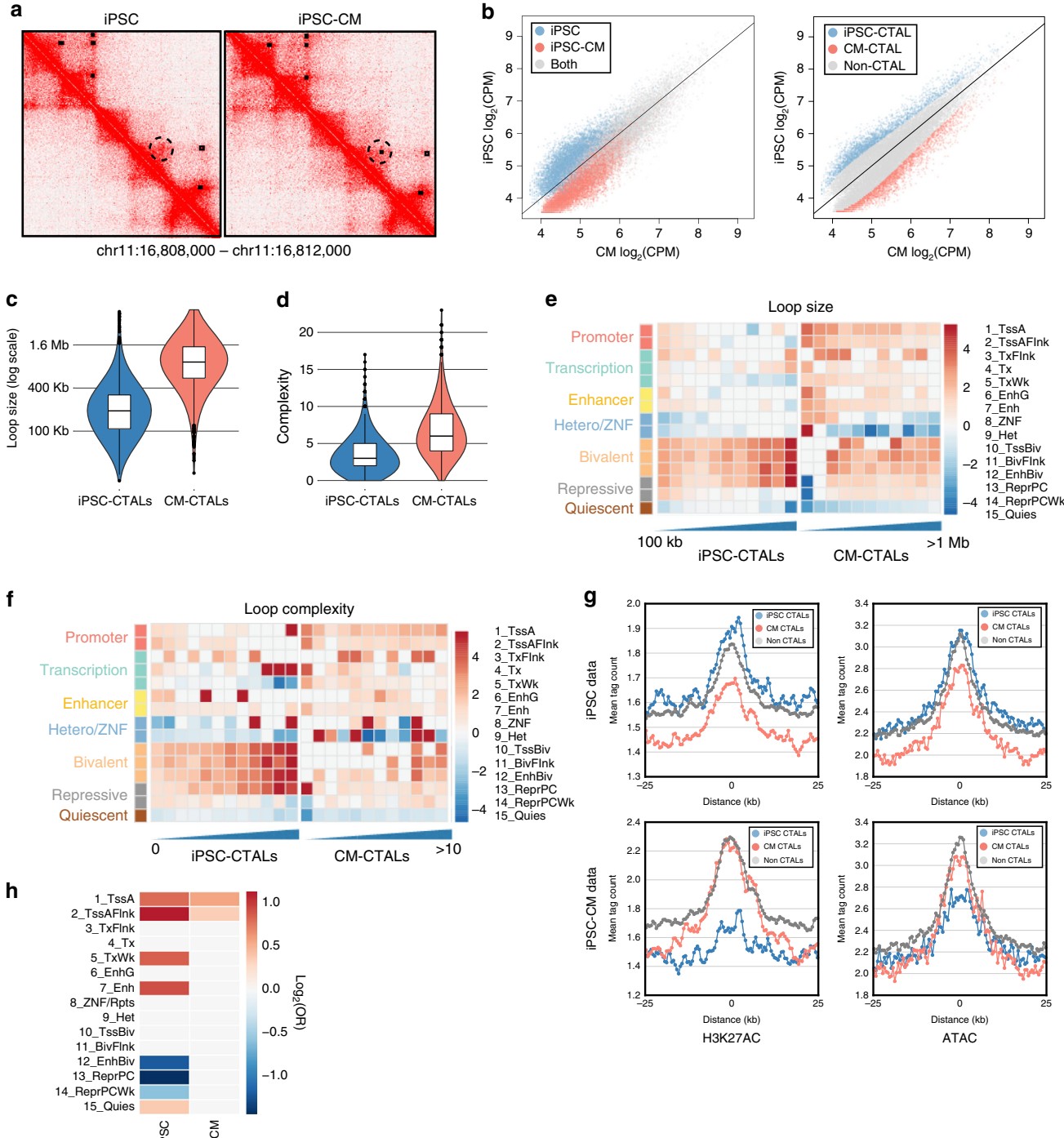

**Fig. 3** Differential chromatin states and sizes in CTALs recapitulate changes in looping across differentiation. **a** Example contact maps showing a loop which appeared in both cell types but was only called in one cell type. Loop calls are shown in the top right half of the contact map as black rectangles. A dotted circle has been added to highlight the region which appears the same in both cell types, but only has a loop call within iPSC-CMs. **b** Scatterplots showing contact frequency in counts per million (CPM) of all loops identified in either iPSCs or iPSC-CMs. The solid black lines indicate the function $y = x$. (left) Points are colored to indicate loops called in only iPSC (blue), only iPSC-CM (red), or both (gray). (right) Points are colored to indicate loops with significantly increased contact frequency in iPSCs (iPSC-CTAL; blue), iPSC-CMs (CM-CTAL; red), or neither (non-CTAL; gray). **c**, **d** Violin plots (all four quartiles shown via lower whisker, lower half of box, upper half of box, and upper whisker; lines indicate median) showing distributions of **c** loop size, and **d** loop complexity for CTALs. **e**, **f** Heatmap showing enrichment of regulatory regions near iPSC-CTAL (left) and CM-CTAL (right) at loop anchor centers with loops stratified by **e** size or **f** complexity. The 15 ROADMAP chromatin states of iPSC (E020) or fetal-heart tissue (E083) were used, and the log₂ odds ratio of enrichment is indicated by color (red positive, blue negative). CTALs broken down by size into **e** 100 kb windows, or **f** complexity. **g** Line plots showing the mean tag intensity of H3K27AC (left) or ATAC-seq data (left) from iPSC (top) or iPSC-CMs (bottom) at iPSC-CTAL (blue), CM-CTAL (red), or non-CTAL (gray) anchors. Cell type data is enriched at cell type CTAL anchors, and non-CTAL anchors, whereas non-cell type data is depleted at cell type CTAL anchors. **h** Heatmap of log₂(odds ratio) from a Fisher's exact tests for enrichments of differential chromatin states across CTAL anchors. White cells indicate a non-significant Fisher's exact test (FDR $q > 0.05$). log₂(OR) is shown by color (red positive, blue negative)

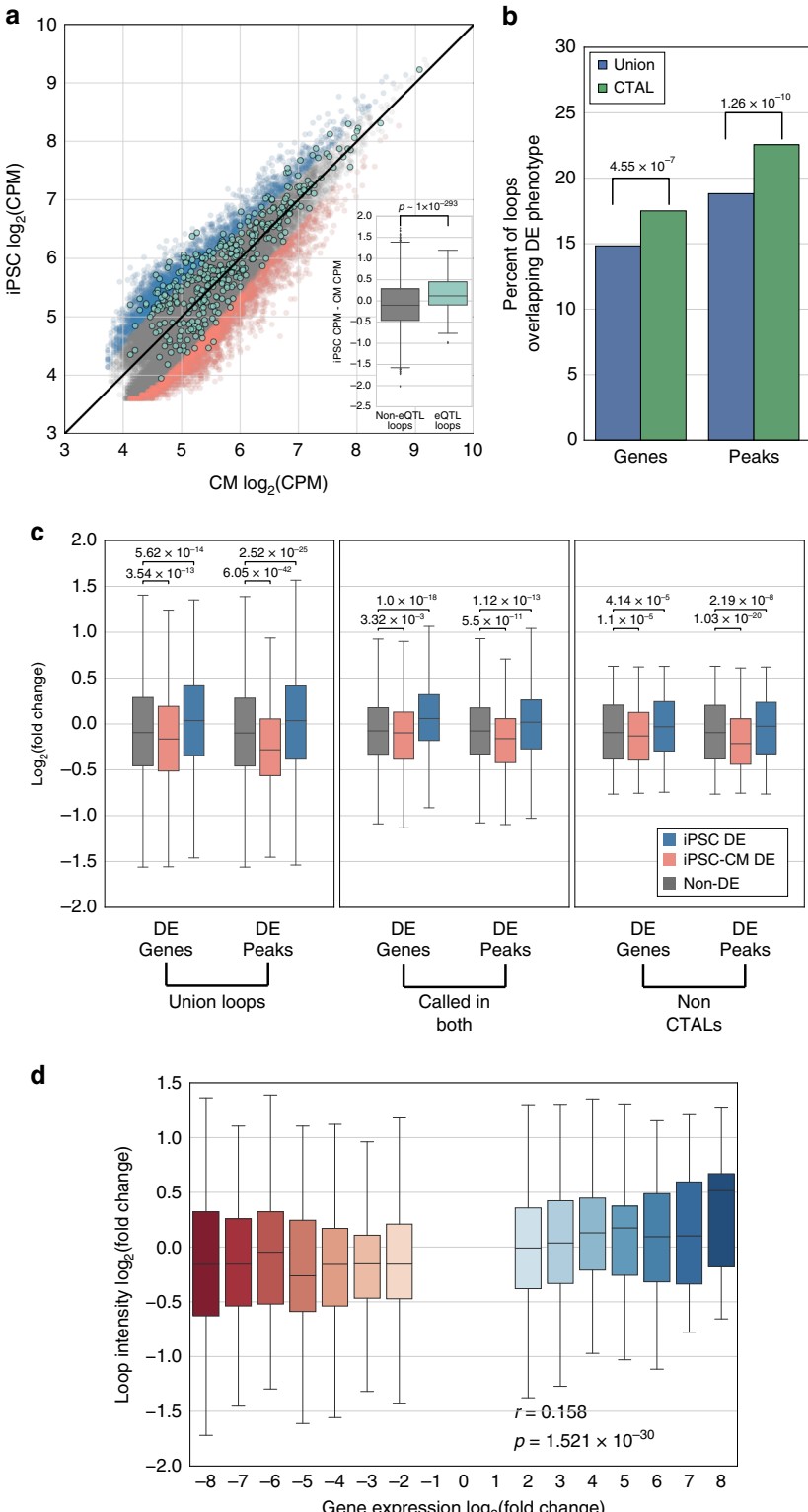

propensity and differential expression, as well as the quantitative association between contact propensity and differential H3K27ac, across cell types. We tested whether the fold change in contact frequency across cell types was in the direction of the cell type with higher differential expression or H3K27ac. We found that across the union loop set, anchors overlapping DE genes with higher expression in iPSCs had significantly greater contact

frequency in iPSCs, and anchors overlapping DE with higher expression in iPSC-CMs had significantly higher iPSC-CM contact frequency; similar patterns were found for DE H3K27ac peaks (Mann–Whitney U test $p < 0.05$; Fig. 4c left). To establish that this association was due to the differences in contact propensity, rather than driven by loops that were differentially called between the two cell types, we examined whether this association

**Fig. 4** Quantitative variation in chromatin loops is associated with differential gene expression and H3K27ac across cell types. **a** Scatter plot showing iPSC vs. iPSC-CM contact frequencies in counts per million (CPM) for all union loops. The black line indicates the $y = x$ function. Background points indicate iPSC-CTALs (blue), CM-CTALs (red), and non-CTALs (gray). Overlaid on this are points indicating iPSC eQTL-eGene containing loops (teal). The boxplot in the lower right corner of the scatter plot shows the fold change between iPSC and iPSC-CM CPMs at non-eQTL loops (gray) or eQTL loops (teal). Positive values indicate a loop had higher CPM in iPSCs, and negative values indicate a loop had higher CPM in iPSC-CM. The p-value was calculated from a Mann–Whitney U test. **b** Barplot showing the percent of CTALs (green) or union loops (blue) which overlap differentially expressed genes or H3K27ac peaks. P-values were found via a Fisher's exact test on the underlying counts of differentially expressed genes or peaks between union loops and CTALs. **c** Boxplots (all four quartiles shown via lower whisker, lower half of box, upper half of box, and upper whisker; lines indicate median; outliers not shown) of the $\log_2$(fold change) of contact frequency at chromatin loops, with positive indicating higher contact propensity in iPSCs and negative indicating higher contact propensity in iPSC-CMs for all loops (i), loops called in both cell types (ii), or non-CTALs (iii) with anchors overlapping differentially expressed genes or H3K27ac peaks with higher expression or counts in iPSCs (blue), higher expression or counts in iPSC-CMs (red), or not overlapping a DE gene or peak (gray). P-values were calculated via a Mann–Whitney U test. **d** Boxplot (all four quartiles shown via lower whisker, lower half of box, upper half of box, and upper whisker; lines indicate median; outliers not shown) showing the $\log_2$(fold change) of chromatin loop frequency for chromatin loops overlapping a differentially expressed gene, binned by the $\log_2$(fold change) of the gene. For both expression and chromatin looping, positive indicates stronger counts in iPSCs, and negative indicates stronger counts in iPSC-CMs. The Pearson correlation and p-value shown were calculated on the raw underlying data

was still present within only the loops that were called in both cell types (i.e., the intersection of iPSC-CM and iPSC called loops). We found that the statistically increased contact frequency (Mann–Whitney U test $p < 0.05$) in the upregulated cell type remained within this set of loops, though the extent of the differences in chromatin looping were smaller (Fig. 4c middle). Thus, we next examined whether these differences could be observed at non-CTALs (i.e., loops with non-significant differences across cell types) and found that these loops were still significantly stronger in the expected direction when they overlapped a DE molecular phenotype at their anchor (Fig. 4c right). These results suggest that subtle variation in chromatin looping across cell types may be functional. Finally, to examine whether chromatin loop contact propensity proportionally varied with the strength of gene expression differences between cell types, we examined the correlation between fold changes in gene expression and chromatin loop contract frequency at loops with anchors overlapping promoters of differentially expressed genes (Fig. 4d). We observed a significant correlation ($r = 0.158$, $p < 1.6 \times 10^{-30}$) between the two phenotypes; however, the magnitudes at which the phenotypes varied were quite different, with gene expression varying up to 250-fold, and the middle three quantiles of chromatin looping varying less than threefold. For these analyses, we pooled data across the genome to measure the association between contact frequency and gene expression, independent of a particular locus; therefore, this analysis compares the relationship between contact propensity and gene expression in aggregate across the genome. As each pair of fold change measurements between contact frequency and gene expression are from the same locus in two different cell types, locus specific biases based on the linear genome which affect Hi-C read depth (number of restriction enzyme sites near the anchors, anchor GC content, and mapping uniqueness)[45] are held constant. Overall, these results suggest that small magnitude changes in contact propensity may be functional as they are associated with large magnitude changes in gene expression across cell types.

**Haplotype-based interrogation of loops and gene regulation.** To enable the functional characterization of haplotype-specific chromatin looping, we phased the Hi-C, H3K27ac, and RNA-seq data to obtain haplotype-associated phenotype data (Supplementary Data 4–14). We first phased the WGS genotype data for these seven individuals using a combination of Hi-C-based phasing and family structure, resulting in an average of 2.01 M phased heterozygous variants per individual (Supplementary Figure 4, see Methods). Next, we assigned informative reads from H3K27ac and RNA expression to each individual's maternal or

paternal haplotype using MBASED[46], and then identified significant peaks or genes with allele-specific effects (ASE; FDR $q < 0.05$) within each individual using a binomial test. We identified a total of 189 ASE peaks (mean 43 per individual) in iPSCs and 618 ASE peaks (mean 119 per individual) in iPSC-CMs, and 2582 ASE genes (mean 647 per individual) in iPSCs and 2214 ASE genes (mean 503 per individual) in iPSC-CMs.

To characterize haplotype-specific chromatin looping, we performed a genome-wide analysis to identify haplotype-associated chromatin loops with consistent significant allelic imbalance (haplotype-associated loops; HTALs) across individuals. Within each cell type, for each individual, we assigned informative Hi-C contacts carrying a phased allele to each haplotype (Fig. 5a) and examined allelic imbalance across all loops. Next, for each individual, we identified imbalance via a Z score using a half-normal distribution (as well as using the computational framework WASP; see Methods and Supplementary Figure 7 for details of complementary analysis), following which we combined the p-values across individuals with Fisher's method for meta-analysis. This process identified 54 total HTALs: 27 from iPSCs, and 27 from iPSC-CMs. We first examined whether these 54 HTALs were enriched for being CTALs of either cell type and found no significant enrichments (Fisher exact $p > 0.05$). We next examined whether the HTALs were truly cell type specific or if the sparsity of the Hi-C data statistically limited our ability to detect allelic loop imbalance present in both cell types. For each of seven individuals, we determined the individual's maternal allele ratio for each of the 27 iPSC HTALs using the iPSC Hi-C data, as well as the maternal allele ratio using the iPSC-CM Hi-C data (Fig. 5b; Supplementary Figure 5A). We then repeated this process at each of the 27 iPSC-CM HTALs (Fig. 5c; Supplementary Figure 5B). For both cell types, we found the maternal allele ratios to be highly correlated with the other cell type across all individuals ($0.73 < $ Pearson's $r < 0.97$), which suggests that loop imbalance was consistent across both cell types. As we observed that the maternal allele frequencies were highly correlated across cell types, to increase power for these analyses, for each of the 26,679 chromatin loops in the union set, we pooled contacts for each individual across their corresponding iPSCs and iPSC-CMs. We observed a median of 50 informative contacts per individual per loop, which corresponds to 100% power to identify HTALs with an allelic imbalance ratio of 70% or higher with $\alpha = 0.02$ in an individual (Supplementary Figure 5C), or at $\alpha = 2 \times 10^{-5}$ when all samples display similar imbalance and are combined with Fisher's method meta-analysis. Within each subject, a mean of 6.08% of all chromatin loops showed significant imbalance at $p < 0.05$ (Z score on a half-normal distribution; see Methods), slightly higher than the statistically expected 5% by chance; however, only a mean of 0.1% (26.6) were

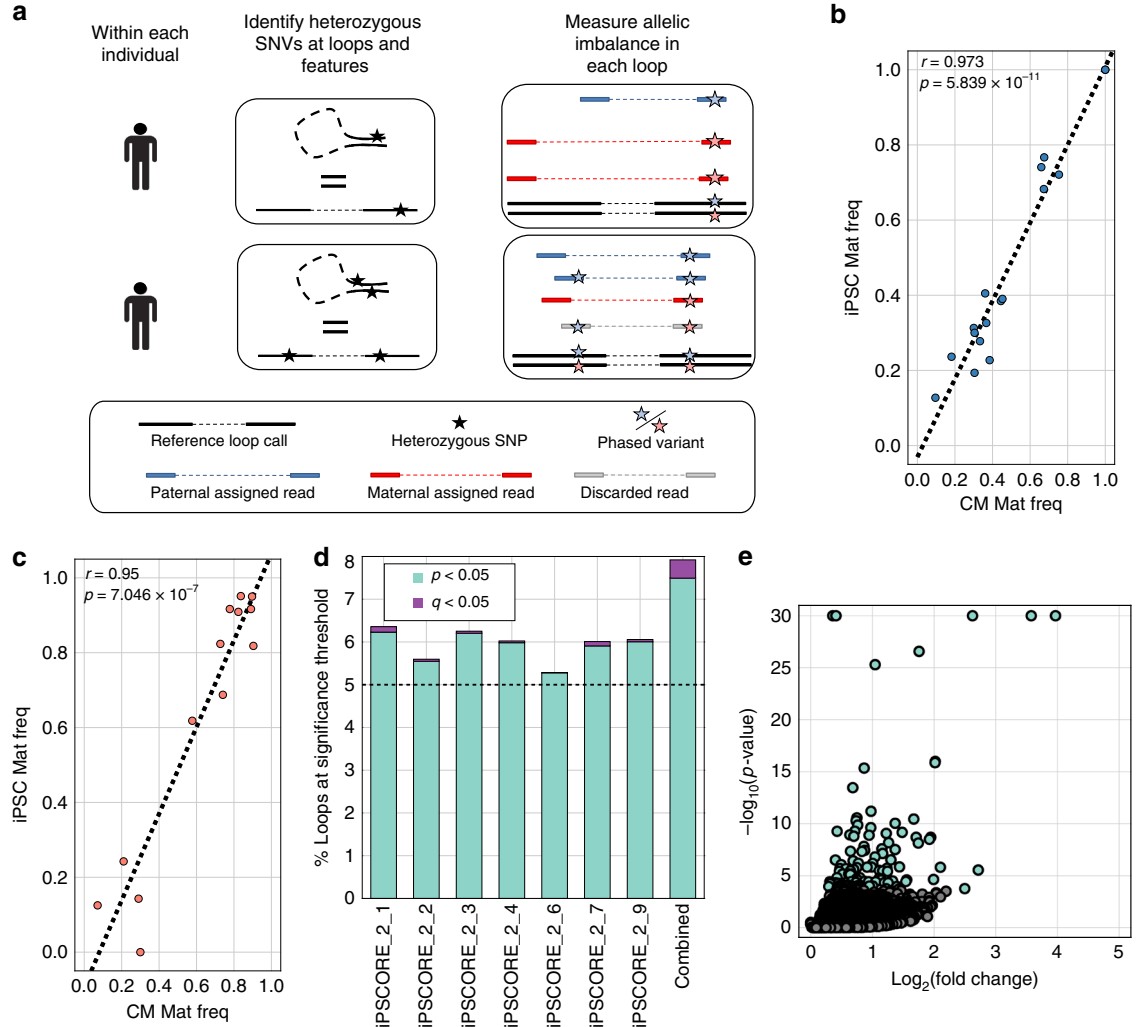

**Fig. 5** Identification of haplotypic differences of chromatin conformation. **a** Schematic showing approach to quantify chromatin loop imbalance within each individual. Examples for two different individuals are shown. Variants were phased using Hi-C and family structure (see Methods), and each contact was assigned to its corresponding haplotype based on the phase of heterozygous SNVs it contained. Reference loop calls and unphased heterozygous SNPs are shown in black. Phased variants are shown in red for maternal, and blue for paternal. Reads only overlapping paternal phased variants were assigned to the paternal haplotype (blue reads), and read only overlapping maternal phased variants were assigned to the maternal haplotype (red reads). If a read overlapped both types of variants, it was discarded (gray read). **b**, **c** Scatter plot showing comparison between iPSC and iPSC-CM maternal haplotype frequencies for one of the seven individuals at HTALs identified in either **b** iPSCs or **c** iPSC-CMs. Linear regression correlation and p-value are reported for each cell type. Similar plots for all seven individuals are in Supplementary Figure 5. **d** Barplot showing the percent of loops associated with haplotype imbalance at $p < 0.05$ shown in teal, and with those also $q < 0.05$ shown in purple. Bars are shown for each individual separately, or for the results of a Fisher's method meta-analysis p-value (Combined; right most bar). A dashed line is drawn at 5% to indicate the number of HTALs expected by chance to be significant at $p < 0.05$. **e** Volcano plot showing the $\log_{10}$(p-value) vs the $\log_2$(fold change of allelic imbalance ratio, with the higher frequency allele always in the numerator of the ratio; see Methods) for each loop with the combined data. As only fold changes of major allele frequencies can be calculated due to haplotypes not having a single reference or alternate allele, all fold changes are positive. Significant points (HTALs) are shown in teal

significant under FDR $q < 0.05$ in each individual (Fig. 5d). To identify HTALs which were consistently imbalanced across individuals, we again combined associations using a Fisher's method meta-analysis for each loop, and identified 7.49% of chromatin loops as HTALs at $p < 0.05$, indicating that consistent allelic imbalance occurs more frequently than by chance. However, only 114 HTALs were significant after multiple testing corrections at FDR $q < 0.05$ (equivalent to $p < 2 \times 10^{-5}$), showing that even with the increased power by using the combined cell type data, the majority of loops had small allelic differences (Fig. 5e). In comparison, we observed slightly fewer HTALs ($N = 89$) with the WASP analysis; however, the majority (83/89, 93%) were found in both sets. These results and power indicate that while we may not detect all small haplotype

differences (i.e., those with imbalance < 70%), large haplotype differences in chromatin looping occur infrequently.

**HTALs are associated with imprinting and CNVs.** We next examined whether the 114 genome-wide significant HTALs were statistically more likely to be a specific type of loop, or overlap genomic features previously shown to be associated with differential chromatin looping (imprinted genes[2,5] and somatic and inherited CNVs[20,47]; Supplementary Table 3). We first hypothesized that chromatin loops that were variable across cell types may be more variable in general, and thus HTALs would be more likely to be CTALs. We compared the proportion of the 114 HTALs that were

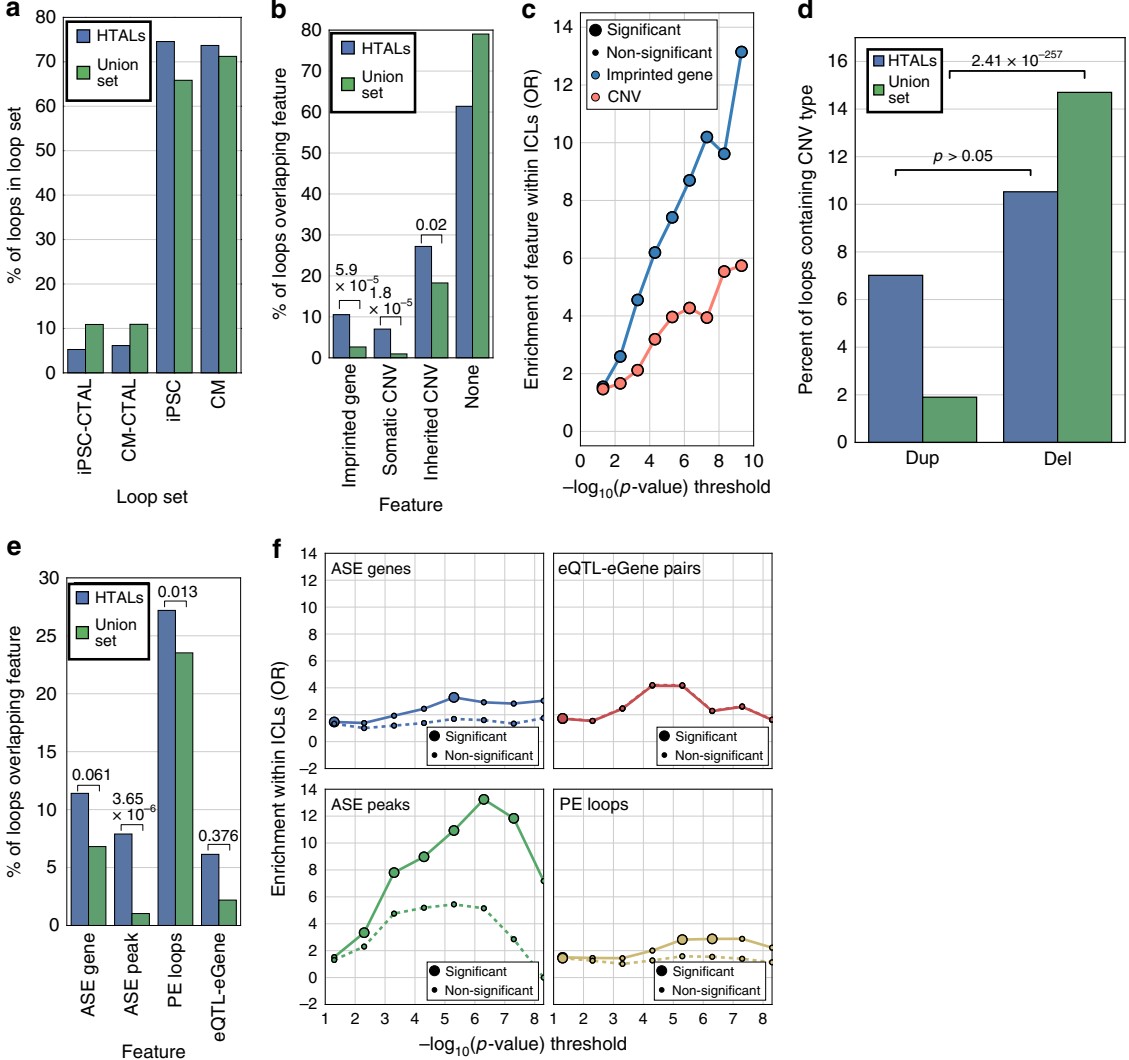

**Fig. 6** Functional characterization of haplotypic differences in chromatin conformation. **a** Barplot showing the percent of union loops (green) or HTALs (blue) contained within each loop set. **b** Barplot showing the percent of union loops (green) or HTALs (blue) containing the given genomic feature within it (i.e., the genomic feature overlapped the region between the start of the first anchor and the end of the second anchor). $P$-values were calculated using via a Fisher's exact test. **c** Line plot showing odds ratio from a Fisher's exact test for HTAL enrichment above the union set for containing an imprinted gene (blue) or containing either an inherited or somatic CNV (red) as a function of the $-\log_{10}$ of the HTAL imbalance $p$-value. Large circles indicate that the test was significant after Bonferroni correction, and small circles indicate a non-significant association. **d** Barplot showing the percentage of union loops (green) or HTALs (blue) containing only deletions or only duplications. $P$-values were calculated using a binomial approximation to a normal distribution, adjusted for the number of identified CNVs which were deletions vs duplications. **e** Barplot showing the percent of union loops (green) or HTALs (blue) overlapping the given genomic feature at an anchor. $P$-values were calculated using a Fisher's exact test. **f** Line plot showing odds ratio from a Fisher's exact test for HTAL enrichment above the union set for containing the labeled feature as a function of the $-\log_{10}$ of the HTAL imbalance $p$-value, for either all loops (solid lines), or loops that do not contain an imprinted gene or CNV (dashed lines). Large circles indicate that the test was significant after Bonferroni correction, and small circles indicate a non-significant association

also iPSC-CTALs, CM-CTALs, iPSC called, or iPSC-CM called loops to the corresponding proportion of union loops. However, we found no significant differences for any association ($p > 0.05$ for all tests; Fig. 6a). We next examined whether a particular type of loop was enriched within HTALs (i.e., CTCF loops, promoter–enhancer loops; see Methods), and found no significant enrichment (FDR $q > 0.05$); together, these results indicate that loops which varied between haplotypes were not more likely to be a specific type of loop. We next compared the distribution of genomic features known to cause large allelic differences within HTALs and the union loop set (Fig. 6b). We observed that, compared to the union loop set, HTALs were statistically more likely to contain imprinted

genes (HTAL: 10.5%; all: 2.7%; Fisher's exact $p = 5.8 \times 10^{-5}$), and somatic (HTAL: 7.0%, all: 1.0%; Fisher's exact $p = 1.8 \times 10^{-5}$) and inherited (HTAL: 27.2%, all: 18.3%; Fisher's exact $p = 2.03 \times 10^{-2}$) CNVs previously identified in these samples[32]. To examine whether these trends held across all levels of imbalance significance, we quantified the extent of association of each genomic feature with chromatin loop allelic imbalance as a function of HTAL $p$-value. For imprinted genes, as the $p$-value threshold increased, the odds ratio increased almost log-linearly, whereas CNV overlap increased but to a lesser extent (Fig. 6c). We next examined the distribution of the types of CNVs contained within loops by examining the subset of loops which contained any number of only a single type of CNV

(Deletion or Duplication, Fig. 6d). While we found deletions to be enriched above duplications within union loops (Binomial $p = 2.41 \times 10^{-257}$), we found no significant enrichment within HTALs. Thus, while CNV type was not associated with allelic imbalance, loop detection may be affected by CNV presence. The observed pattern of enrichment in deletions is consistent with linearly closer loci having increased Hi-C contact propensity (as deletions reduce the linear space between loci) thereby increasing contact frequency and loop detection power; conversely, duplications increase linear distance and thus decrease contact frequency and loop detection power. Thus, it is unclear how much of this enrichment is due to a technical artefact induced by increased power at deletions. Overall, these results confirm previous reports which suggested that genetic imprinting[2,5] may be a strong driver allelic imbalance, and suggest that CNVs may have smaller effects on allelic imbalance in chromatin looping.

**Regulatory genetic variants and contact propensity.** We next examined whether HTALs were enriched for functional allele-specific differences by quantifying the enrichment for containing an ASE gene or ASE H3K27ac peak at their anchors, or for being a promoter–enhancer (PE) or eQTL-eGene loop. We found ASE peaks to be enriched at HTAL anchors, and also being a PE loop to be enriched (Fisher's exact $p < 0.05$; Fig. 6e). Notably, despite the increased percentage of eQTL-eGene loops in HTALs, as only 7 eQTL-eGene loops were HTALs (585 eQTL-eGene loops in total), this increase was non-significant. To determine whether regulatory genetic variation was associated with these differences, we excluded the effects from imprinting and CNVs, and examined these associations across a range of imbalance thresholds (Fig. 6f). The removal of imprinted regions and CNVs greatly attenuated the association, and resulted in a loss of significance for the two molecular phenotypes and PE loop status over almost all ranges of imbalance significance. These results suggest chromatin loops vary across haplotypes much more subtly (i.e., allelic ratio < 70%) than gene expression or H3K27ac, and where variation is larger, it is mainly driven by imprinting and/or CNVs. Additionally, these results show that large allelic imbalances in chromatin loops are primarily restricted to those located in imprinted regions or associated with copy-number variation, and that regulatory genetic variants are not associated with large changes in contact propensity.

**Haplotypes, contact propensity, and gene regulation.** As we observed that subtle differences in contact propensity were quantitatively associated with large differential regulation of gene expression across cell types (Fig. 4d), we investigated if similar small-scale changes in contact propensity across haplotypes were associated with gene expression and regulation differences. We first compared the general variability of chromatin loops (excluding imprinted regions and CNVs) across cell types (Fig. 7a) to the variability across haplotypes (Fig. 7b). We found

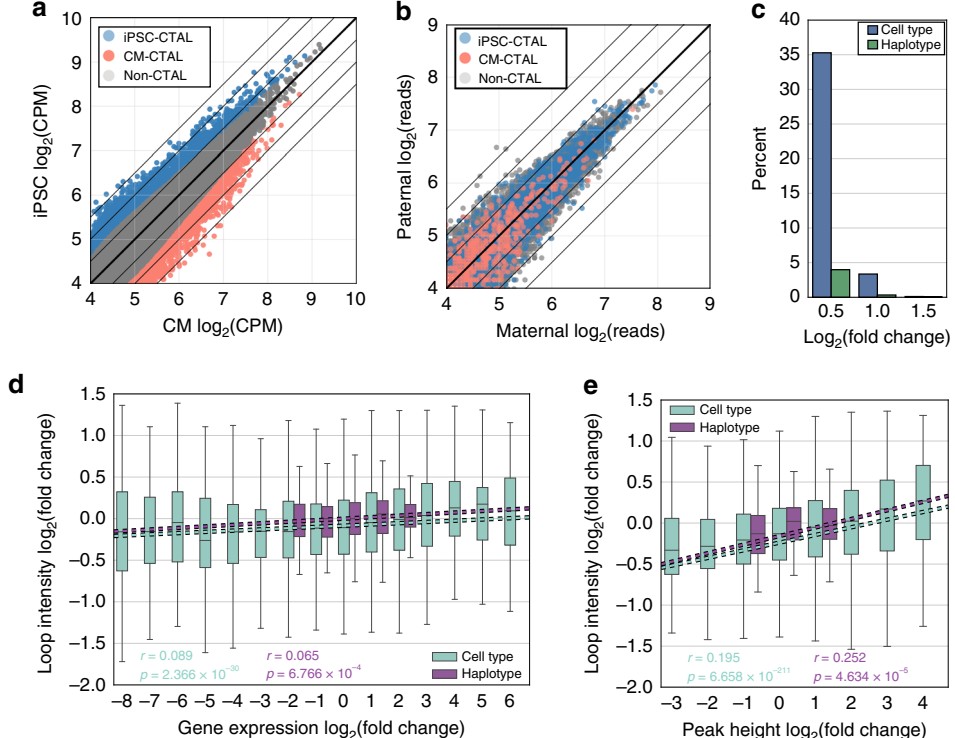

**Fig. 7** Comparison of chromatin loop, gene expression, and H3K27ac variability across cell types and haplotypes. **a**, **b** Scatterplots showing **a** contact frequency in log$_2$ counts per million (log$_2$(CPM)) across cell types or **b** read counts across haplotypes of all union loops colored by CTAL status. The solid bold line indicates the function $y = x$, and other lines indicate absolute fold changes of log$_2$(0.5), log$_2$(1), and log$_2$(1.5). **c** Percent of loops with at least the shown log$_2$(fold change) or across cell types (blue) or haplotypes (green). **d**, **e** Boxplot (all four quartiles shown via lower whisker, lower half of box, upper half of box, and upper whisker; lines indicate median; outliers not shown) showing the log$_2$(fold change) of chromatin loop contact frequency for chromatin loops overlapping a **d** differentially expressed or ASE gene, or **e** differential or ASE H3K27ac peak, binned by the log$_2$(fold change) of the **d** gene or **e** peak. Boxes are shown for cell type comparisons in teal, and haplotype comparisons in purple, linear regressions are plotted with dashed lines, and r's and p-values are shown and colored from the raw data in each dataset independently. For all data, positive fold change indicates stronger counts in iPSCs, and negative fold change indicates stronger counts in iPSC-CMs

that more chromatin loops varied to a larger degree across cell types than across haplotypes: ~35% of loops exhibited a $\log_2$ fold change of 0.5 (1.4-fold) or higher across cell types, whereas only ~5% of loops showed a similar fold change across haplotypes (Fig. 7c). This result suggests that haplotype-associated differences are considerably smaller than cell-type-associated differences. We therefore examined whether the association between contact propensity and gene expression, or contact propensity and H3K27ac, was significant and proportionally consistent across cell types and haplotypes. Across cell types and haplotypes, we found a positive and highly significant correlation (Pearson correlation; cell type: $p = 2.36 \times 10^{-30}$, haplotype $p = 6.76 \times 10^{-4}$, Fig. 7d) between gene expression fold change and chromatin loop fold change, and between H3K27ac fold change and loop fold change (Pearson correlation; cell type: $p = 6.6 \times 10^{-211}$; haplotype: $p = 4.63 \times 10^{-5}$, Fig. 7e). Similar to the cell type analyses (Fig. 4d), we found the range at which gene expression and H3K27ac fold changes occurred to be larger than the range at which loop fold changes occurred. These consistent associations between the cell type and haplotypes analyses, as well as the consistent magnitude differences between looping and molecular phenotype, suggest that large differences in gene expression and H3K27ac are associated with small differences in chromatin loop contact propensity. Additionally, as the association between gene expression and contact propensity was consistent across haplotypes, these results suggest that genetic variation could exert effects on gene expression through small modulation of contact propensity.

## Discussion

Here, we generate a resource of phased genotypes, Hi-C, and molecular phenotype data in two cell types for seven individuals who are a part of a three-generation family, and use this data to perform an in depth, genome-wide, functional examination of changes in contact propensity across cell types and haplotypes. These chromosome-length haplotypes, and accompanying phased data, will enable future studies examining long-range interactions between multiple genetic variants on the same chromosome. Additionally, these Hi-C maps are the highest-resolution maps for human iPSCs and iPSC-CMs currently available and are thus an important resource for the prioritization of functional variants and their potential gene targets in these cell types.

We performed quantitative comparisons of contact frequency across cell types and haplotypes to identify differences in chromatin looping, and integrated these differences with quantitative measures of differential expression and H3K27ac to examine the functionality of contact propensity. These analyses revealed a proportional association between contact frequency and gene expression/H3K27ac, which surprisingly linked the phenotypes across different magnitudes of variability: extremely subtle changes in contact frequency were associated with large differences in gene expression and H3K27ac. If contact propensity at loops is a fundamental regulator of gene expression, differences in contact propensity would be expected to be associated with similarly sized differences in gene expression regardless of the environment in which the differences occurred (i.e., across cell type, haplotype, or experimental conditions). As we observed a consistent relationship between the two, we believe these data indicate that contact propensity is a mechanism involved in regulating gene expression, similar to enhancer activity or transcription factor binding strength. Notably, as we identified a non-directional correlation, contact propensity may either affect, or be affected by, gene expression and/or regulation.

While the mechanisms underlying changes in contact propensity are currently unknown, there are several reasonable hypotheses. Previous studies showing that the physical 3D structure of the genome can be reconstructed from contact frequency via polymer physics models[26,48–51] suggest that differential contact propensity could result from changes in spatial proximity. The fact that CTCF and Pol2 ChIA-PET show similar profiles to Hi-C data[5] suggest that differences in protein binding near loop loci could also affect contact propensity. Finally, as we found associations between contact propensity and H3K27ac, regulatory chromatin activity could modulate contact propensity. Future studies examining these mechanisms could provide insights into the biological processes underlying differential contact propensity and gene regulation.

The identification of specific causal variants associated with differential contact propensity is likely to be challenging, as we did not find a large number of HTALs with strong effects outside of imprinted and copy-number variable regions. As the effects of imprinting are parental in nature, rather than genetic, it is necessary to search outside of these regions for causal regulatory variants. In non-imprinted regions, if we interpolate the association between gene expression and contact propensity, the linear model would suggest that a gene with 98% ASE would be expected to be associated with a loop imbalance of only ~52%. This minute difference in loop imbalance provides a possible explanation for why we did not observe HTALs associated with gene regulation or ASE, but found a quantitative association between Hi-C signal and functional phenotypes overall. Additionally, it suggests that high coverage would be needed to identify HTALs outside imprinted regions. Thus, for the validation of specific variants, or identifying loop QTLs, future studies should consider using an unbiased targeted loop capture assay with higher sensitivity and targeted coverage than Hi-C, such as sequence-based pHi-C, and perform quantitative analyses using these data.

Finally, our work provides some insight into the ongoing question of whether changes in chromatin looping cause changes in gene expression, or if changes in gene expression cause changes in looping[1,2,27,35,43,52–55]. It has been established that the creation of new chromatin loops can alter gene expression[56], however, is has been less clear whether altering gene expression results in meaningful changes in chromatin loops[35,52,57]. Evaluating whether chromatin loop changes are meaningful requires an understanding of the scale at which functional changes in chromatin loops occur. As our findings suggest that subtle changes are functional, we believe these discordant interpretations could have arisen from studies either not being sufficiently powered to detect small effects, or from discounting small changes as nonfunctional. Our work therefore provides a foundation for future studies to quantitatively examine how changes in contact propensity elicit changes in expression (or vice versa) and suggests that studies designed to detect small magnitude changes in chromatin loop variability may be needed to delineate the relationship between chromatin loop imbalance and gene expression.

## Methods

**Subject enrollment**. The seven individuals used in this study were recruited as part of the iPSCORE project[28]. We have complied with all relevant ethical regulations for work with human participants, and informed consent was obtained. iPSCORE recruitment was approved by the Institutional Review Boards of the University of California, San Diego and The Salk Institute (Project no. 110776ZF), and consent forms were received from each subject. Subject information including sex, age, and ethnicity were collected during recruitment (Supplementary Table 1). Skin biopsy was performed to obtain fibroblasts for iPSC reprogramming, and blood samples were collected for whole-genome sequencing.

**iPSC derivation and iPSC-CM differentiation**. Cell line derivation and differentiation were performed as described in Benaglio et al.[31]. From the seven

individuals, fibroblast samples from skin biopsies were reprogrammed using non-integrative Cytotune Sendai virus (Life Technologies)[29] following the manufacturer's protocol. Each independent reprogramming resulted in one or more iPSC clones of the subject. At passages 12–13, genomic integrity of at least one iPSC clone per subject was assessed using Illumina HumanCoreExome arrays, and pluripotency of iPSCs was assessed for most clones in this study by flow cytometry of the pluripotency markers SSEA4 and TRA-1-81[28]. iPSCs of each clone were harvested between passages 12 and 40, resulting in a total of 38 iPSC samples used in this study (Supplementary Data 15). Each iPSC clone was then used to generate multiple independent iPSC-CM differentiations using a monolayer protocol[30], resulting in a total of 27 iPSC-CM samples used in this study. Among these iPSC-CM samples, 11 of them were subjected to purification via 4 mM sodium L-lactate at day 15 after the start of differentiation and collected at day 25[58]; one iPSC-CM sample was subjected to lactate purification at day 11 and collected at day 16; the rest of the iPSC-CM samples were not subjected to lactate purification and collected at day 15 (Supplementary Data 15). Across all molecular assays detailed below, lactate purified and non-lactate purified iPSC-CM samples showed similar profiles; we therefore combined data across the two protocols. Single-nucleotide variants (SNVs) and copy-number variants (CNVs) of these individuals were obtained from ~40× WGS from iPSCORE[28] through dbGAP and phs001325.v1.p1, and from DeBoever et al.[32].

**Hi-C data generation.** For each of the 11 iPSC and 13 iPSC-CM Hi-C samples, we performed in situ Hi-C on 2–5 million cells. Hi-C libraries were prepared using in situ Hi-C[2]. Cells were crosslinked at a final concentration of 1% formaldehyde and quenched using 200 mM glycine. Crosslinked cells were then lysed and nuclei were digested with 100 U MboI overnight at 37 °C. Next, fragmented ends were biotinylated for 90 min at 37 °C, and the sample was diluted and proximity ligated for 4 h at room temperature. Crosslinks were reversed by the addition of SDS, ProteinaseK, and NaCl, and allowed to incubate overnight at 68 °C. Samples were then purified by ethanol precipitation, resuspended in 100 μL 1× Elution Buffer, fragmented using a Covaris S2 instrument, and size selected using AmpureXP beads. Subsequently, biotinylated ligation junctions were pulled down using T1 Streptavidin beads. Hi-C libraries were prepared using streptavidin beads by performing end-repair, dA-tailing, and adapter ligation, following which PCR amplification and purification was performed. The resulting libraries were sequenced on an Illumina HiSeq 4000 machine to obtain 150 bp paired-end reads.

**RNA-seq data generation.** RNA-seq data was obtained from Benaglio et al.[31]. Specifically, total RNA was isolated using the Qiagen RNAeasy Mini Kit from frozen RTL plus pellets, including on-column DNAse treatment step. RNA was eluted in 60 μl RNAse-free water and run on a Bioanalyzer (Agilent) to determine integrity. Concentration was measured by Nanodrop. Illumina Truseq Stranded mRNA libraries were prepared and sequenced on HiSeq2500, to an average of 40 M 100 bp paired-end reads per sample. RNA-seq reads were aligned using STAR[59] with a splice junction database built from the Gencode v19 gene annotation[60]. Transcript and gene-based expression values were quantified using the RSEM package (1.2.20)[61] and normalized to transcript per million bp (TPM).

**ChIP-seq data generation and peak calling.** H3K27ac data was obtained from Benaglio et al.[31]. For H3K27ac, $2 \times 10^6$ fixed cells were lysed in 60 μl of MAGnify™ Chromatin Immunoprecipitation System Lysis Buffer (Thermo Scientific) and sonicated using Bioruptor 200 (Diagenode) for 35–45 min of 30 s on/30 s off cycles. H3K27ac antibodies (Abcam ab4729, lots GR183922-2 (1.75 μg) or GR184333-2 (1 μg)) were coupled for 2 h to ProteinG Dynabeads (Thermo Scientific), and used for overnight chromatin immunoprecipitation in IP buffer (1% Triton-X, 0.1% DOC, 1× TE, 1× Roche Complete Proteinase Inhibitor tablets (RCPI)). Beads were washed five times with washing buffer (50 mM Hepes pH 8, 1% NP-40, 0.7% DOC, 0.5 M LiCl, 1 mM EDTA, and 1× RCPI) and once with TE buffer. DNA was eluted and reverse crosslinked overnight in elution buffer (10 mM Tris-HCl pH 8, 1 mM EDTA, 1% SDS) at 65 °C. DNA was purified using Qiagen MinElute PCR Purification kit, quantified by Qubit (Thermo Scientific) and submitted to library preparation and barcoding using KAPA Hyper Library preparation kit (KAPA Biosystems). Libraries were sequenced on an Illumina HiSeq2500 or a HiSeq4000 to an average of 35 M 100 bp paired-end reads per sample.

ChIP-seq reads were mapped to the hg19 reference using BWA[62]. Duplicate reads, reads mapping to blacklisted regions from ENCODE, reads not mapping to chromosomes chr1-chr22, chrX, chrY, and read pairs with mapping quality Q < 30 were filtered. Peak calling was performed using MACS2[63] ('macs2 callpeak -f BAMPE -g hs -B --SPMR --verbose 3 --cutoff-analysis --call-summits -q 0.01') using pooled BAM files from all iPSC or iPSC-CM samples and with reads derived from sonicated chromatin not subjected to IP (i.e., input chromatin) from a pool of samples used as a negative control.

**ATAC-seq data generation and peak calling.** ATAC-seq data was obtained from Benaglio et al.[31]. Specifically, the ATAC-seq protocol has been adapted from Buenrostro et al.[64]. Frozen nuclear pellets of $5 \times 10^4$ cells each were thawed on ice, suspended in 50 μL transposition reaction mix (2.5 μL Tn5 transposase in 1× TD buffer, Illumina Cat# FC-121-1030), and incubated for 30 min at 37 °C. Reactions

were purified using Qiagen MinElute kit, eluted in 10 μL water and amplified using the KAPA real-time library amplification kit (KAPA Biosystems) with barcoded adaptors. PCR reactions were terminated after 10–13 cycles and purified using AmPure XP beads (Beckman Coulter). Samples were size selected using SPRIselect beads (Beckman Coulter) to a size range of 150–850 kbp and sequenced on an Illumina HiSeq2500 to an average depth of 30 M 100 bp paired-end reads.

ATAC-seq reads were aligned using STAR to hg19 and filtered using the same protocol as for ChIP-seq. In addition, to restrict the analysis to regions spanning only one nucleosome, we required an insert size no larger than 140 bp, as we observed that this improved sensitivity to call peaks and reduced noise. Peak calling was performed using MACS2 on merged BAM files of iPSC and iPSC-CM meta-samples with the command 'macs2 callpeak --nomodel --nolambda --keep-dup all --call-summits -f BAMPE -g hs', and peaks were filtered by enrichment score (q < 0.01).

**Creation and analysis of Hi-C contact maps.** For each sample, Hi-C reads were first aligned to human reference genome hg19 using BWA-MEM (version 0.7.15)[62] with default parameters. Forward and reverse reads from the paired-end data were aligned independently to allow for identification of split reads that represent ligations between two genomic loci due to spatial proximity[2]. Paired-end reads were then reconstructed, processed, and filtered using the Juicer pipeline[65], resulting in the removal of unmapped reads, abnormal split reads (split reads that cause ambiguous positioning of the contact), read pairs within the same restriction enzyme fragment, low mapping quality read pairs (MAPQ < 30), and duplicate reads. Subsequently, read pairs that were less than 2 kb apart were removed to avoid self-ligated fragments. These filtered read pairs (contacts) were subsequently used to generate chromatin contact maps for each sample via Juicer. To create Hi-C contact maps on a per individual basis, contacts were pooled across all samples of a particular cell type for each individual, and to create maps of iPSC and iPSC-CM, contacts were pooled across individuals within the respective cell type. These processes resulted in a set of binary.hic files, which were utilized to obtain raw and Knight-Ruiz (KR)[66] normalized counts as well as normalization vectors of contact frequency matrices via Juicebox command line tools[67] at various resolutions used throughout this study.

**Correlation of Hi-C contact maps between samples.** The KR normalized contact matrices of each sample were retrieved from the.hic files at 1 Mb using Juicebox[67]. The contact matrices were then vectorized in order to calculate Pearson correlation between each of the samples in R. Hierarchical clustering analyses of the Pearson correlation were performed in R using hclust with default settings and (1 − Pearson correlation) as dissimilarity height. HiCRep was run using the default parameters on chromosome 22 as suggested by the documentation[33].

**Identification of chromatin loops.** Chromatin loops in iPSC and iPSC-CM were called using both Fit-Hi-C[68] and HICCUPS[2,67] as summarized in Supplementary Figure 6A. For Fit-Hi-C, loops were called in meta-fragment resolutions that each contained a fixed number of consecutive restriction enzyme (RE) fragments, ranging from 10 to 30 RE fragments. Loop calling procedures for each resolution are summarized in Supplementary Figure 6B. First, significant interactions (FDR q < 0.01) were identified through jointly modeling the contact probability using raw contact frequencies and KR normalization vectors with the Fit-Hi-C algorithm (Step 1). Next, the output of Fit-Hi-C was pruned by requiring that: (1) the interaction itself was significant; and (2) for each anchor of the interaction, three of the five immediately upstream or downstream bins from the opposing anchor were significant (Step 2). We then merged high-confidence interactions within 20 kb using pgltools[69] (Step 3), discarded interactions that did not have any other interactions within 20 kb, and retained the most significant call at each interaction event (Step 4).

For HICCUPS, loops were called using fixed-size bin resolutions from 5 to 25 kb at 1 kb bin-size intervals using parameters summarized in Supplementary Table 4. Briefly, default parameters of peak size ($p$) and window size ($i$) were used to call loops at 5 and 10 kb resolutions provided by HICCUPS[67], and parameters for other resolutions were chosen by linearly scaling the parameters with respect to the resolution chosen. Specifically, for 6, 7, 8, and 9 kb resolutions, the values of these two parameters were interpolated from the 5 and 10 kb values, and rounded to the closest integer. For resolutions >10 kb, the default 10 kb parameters were used. Following loop calling, as performed by Rao et al.[2], for resolutions from 5 to 10 kb, loops within 20 kb were merged using pgltools. For resolutions above 10 kb, loops within twice the size of the anchor were merged using pgltools. At each merging event, the loop call with the most statistical significance provided from HICCUPS output was retained.

Loop calling techniques are known to be technically variable[34]. We found many loop calls from both Fit-Hi-C and HICCUPS that were located at random points throughout the Hi-C matrix far off the diagonal (Supplementary Figure 1C). We thus developed a procedure to remove these loop calls by examining the number of resolutions at which the loop was identified. We intersected loop calls across all resolutions within each calling method, retaining the highest-resolution call at each intersection event, and filtered out loops present in less than three or seven resolutions for HICCUPS or Fit-Hi-C, respectively. The loops retained in these

filtered sets visually appeared to best represent the underlying Hi-C data (Supplementary Figure 6C). Next, we compared how these filtered sets overlapped with promoter-capture HiC[70] or the Rao et al. loop set and found that using these filtering criteria resulted in a higher overlap with the retained loops (Supplementary Figure 6D), suggesting that this filtering strategy removed spurious loop calls. After this filtering, while we found a large number of loops that overlapped between Fit-Hi-C and HICCUPS (Supplementary Figure 6E), many loops were unique to only one caller (Supplementary Figure 6F). We therefore intersected the loops across calling methods, retaining the loop with the smallest total anchor size at each intersection event (Supplementary Figure 6G). Overall, this process retained the smallest resolution loop call for all loops present in either three HICCUPS or seven Fit-Hi-C resolutions, and resulted in the iPSC called and iPSC-CM called loop sets.

**Identification of TADs.** To identify TADs, we utilized the HMM method from Dixon et al.[10] with the Hi-C matrix at 40 kb resolution as recommended. To determine the percent of loops that were at TAD boundaries, we paired TAD boundaries sequentially in the file to create a pgl format file, and then used pgltools intersect to find the percent of loops with both anchors at TAD boundaries.

**Identification of compartments.** Chromatin compartments were called for each cell type via Juicer command line tools using the corresponding .hic files where the first PC of the normalized contact frequency matrices were extracted at 1 Mb resolution. The signs of the PC eigenvectors were used to stratify each chromosome into two arbitrary compartments. To determine the activity status of the two compartments on each chromosome, we counted the number of reads from (1) RNA-seq, (2) H3K27ac ChIP-seq, and (3) ATAC-seq aligned to each of the 1-Mb bins from all available samples for each cell type, averaged the read counts across all samples for each assay in each cell type, and assigned the compartment with higher average read counts from all the three assays as the active compartment (A) and the other compartment as inactive compartment (B). While most of the time all three assays had consistent compartment activity calls, chr21 of iPSC and chr22 of iPSC-CM had inconsistent calls, where we assigned the compartment activity based on the majority of assays.

**Creation of the union loop set.** To create the union loop set, we used pgltools merge to find all loops from the iPSC call set and iPSC-CM call set with both anchors within 20 kb. This process led to merge events of 1, 2, or 3 loops, which were resolved as follows: (1) if there was only one loop present within 20 kb (i.e., only one loop set had a call), this loop was retained, (2) if there were two loops present within 20 kb, the loops were merged by pgltools merge, (3) if there were three loops present, pgltools closest was used to identify which two loops were closest together; these two loops were merged, and the third loop was retained as its original call.

**Identification of CTALs.** After filtering contacts with Juicer, raw contact frequencies for union loops were obtained by intersecting the filtered read pairs from the 11 iPSC and 13 iPSC-CM Hi-C samples with the union loop set using pgltools coverage. These raw contact frequencies were used as input in edgeR[42], normalized to remove library size bias using trimmed mean of M values (TMM), and compared between the 11 iPSC and 13 iPSC-CM samples using quasi-likelihood F-test. By comparing Hi-C read coverages at the same genome loci in two cell types, the linear genome biases that are known to affect Hi-C are held constant (restriction enzyme cut site frequency, GC content, and mappability)[45]. The significant differential loops were determined by FDR adjusted $q < 0.01$.

**Creation of null loop sets for functional comparisons.** As chromatin loops, and genome annotations such as chromatin states, are highly structured and depend on genomic distance both between their own anchors and other chromatin loops, we used permutation to test for functional enrichment within chromatin loops and at loop anchors. We generated 1000 null loop sets for both the iPSC called and iPSC-CM called loop sets to use for statistical analysis, as genome-wide background levels of genomic traits may not accurately represent a true random distribution of paired-genomic loci. The null loops were generated for each chromosome by (1) removing the gap regions on the human reference genome obtained from UCSC genome browser (https://genome.ucsc.edu/) and updating the loop positions according to this no-gap-genome; (2) sliding the loop positions on the no-gap-genome for a consistent random distance $d$ such that $2\,Mb < d < $ chromosome size - 2 Mb for each null set; and (3) gap regions were added back to the genome, null loop positions were updated back to hg19. In step 2, when loop positions moved beyond the chromosome size after rotation, loops were instead moved to the beginning of the chromosome. Null loops with anchors overlapping a gap region were removed (an average of 0.5% loops were removed in each cell type).

**Distribution of motifs and tag frequencies at anchors.** The findMotifsGenome. pl script from HOMER (v4.7) was used to determine enriched motifs at loop anchors, using the entire size of the anchor as the search space. The HOMER script annotatePeaks.pl was used to identify the distribution frequencies of CTCF motifs,

H3K27ac ChIP-seq reads, or ATAC-seq reads in each set of loop anchors with a bin size of 500 bp and a window size of 50 kb using all bam files for the respective molecular phenotype simultaneously.

**Determining enrichment of chromatin states at loop anchors.** For each of the ROADMAP tissues[71], the core 15-chromatin-state models were obtained as BED format from http://egg2.wustl.edu/roadmap/web_portal/chr_state_learning. html#core_15state, and the states were separated into their original 200 bp bins. To determine the enrichment of each chromatin state at a loop anchor, we compared the proportion of 200 bp bins in the state of interest on the loop anchor, to the genome-wide background level of the bins via Fisher's exact test. A significance level of $p < (0.05/15)$ was considered significant.

**Identification of differential peaks and genes.** To identify differential H3K27ac peaks and genes, we first used featureCounts[72] to obtain the number of reads for each assay from each gene as annotated in gencode v19, or from each peak identified by merging all the H3K27ac data together. Next, we used DEseq2 v1.10.1[73] with default parameters to identify differential peaks and genes with a $\log_2$(fold change) > 2 and an FDR corrected $q$-value < 0.05.

**Enrichment of cell type specific regulatory regions at CTALs.** To determine if cell type specific regulatory regions were enriched at CTALs, for each cell type, we first split the union loop set into CTALs and non-CTALs. Next, we examined whether the proportion of CTALs overlapping a cell type specific regulatory region was statistically larger than the proportion of non-CTALs. For example, to test whether iPSC-CTALs were more likely to harbor an iPSC-specific active promoter, we restricted the analysis to loops overlapping an iPSC active promoter, and tested whether the proportion of loops overlapping an iPSC-specific active promoter was higher within CTALs than non-CTALs. For all analyses, we used Roadmap E020 (iPSC) for iPSCs, and Roadmap E083 (fetal heart) for iPSC-CMs. We defined an anchor as overlapping a cell type specific regulatory region as an anchor which overlapped the region in the tested cell type (E020 for iPSC-CTALs and E083 for CM-CTALs), but did not overlap the region in the other cell type (E083 for iPSC-CTAL comparisons, E020 for CM-CTAL comparisons).

**Phasing genomes.** To obtain accurately phased genotypes for each sample, we performed initial phasing using the Hi-C data, and then subsequently utilized family structure to identify, and fix or remove, haplotyping errors (point errors). We first determined the initial phased genotypes for each individual, at each site at least one individual was heterozygous, by analyzing the HiC data with Haploseq[74]. Next, as Haploseq only identifies heterozygous sites, we filled in missing genotype data with unphased genotypes from iPSCORE WGS variant calls for these individuals (Supplementary Figure 3A). To determine the corresponding parental haplotype for each child haplotype (parent–child haplotype combination), we identified the average concordance between each child haplotype, and each of the four parental haplotypes, in 1 Mb bins chromosome by chromosome, and identified the best matching parent–child haplotype combination for each child chromosome. Within each parent–child haplotype combination, we identified meiotic recombinations within the parent so that we could identify and fix point errors across the genome (Supplementary Figure 3B). We identified recombinations by finding the extreme points from the following scoring function: for a given child haplotype C1, haplotypes from a single parent PH1 and PH2, and $N$ heterozygotic sites across the genome in the child,

$$\text{Score} = \sum_{i=1}^{N} \begin{cases} 1 \text{ if } C_i = PH1_i \text{ and } C_i \neq PH2_i \\ -1 \text{ if } C_i = PH2_i \text{ and } C_i \neq PH1_i \\ 0 \text{ otherwise} \end{cases} \tag{1}$$

We then split each parent–child haplotype combination into crossover blocks at each crossover position so that each child SNV could be compared to both matching parental haplotypes simultaneously, and fixed switch errors according to Mendelian inheritance. Additionally, if any member of the family was unphased at the site, we phased these variants to follow Mendelian inheritance, generating switch error free genotypes (Supplementary Figure 3C). After phasing each trio individually, we re-evaluated Mendelian inheritance across all seven individuals, and removed any sites where Mendelian inheritance was violated, as these indicated genotyping errors in one or more individuals.

**Identification of genome-wide imbalanced chromatin loops.** To identify haplotype-associated chromatin loops (HTALs), we phased contacts from each chromatin loop in the union loop set across cell types, and identified allelic imbalance that was statistically significant at a genome-wide threshold. We first identified all contacts within 25 kb of a loop, kept those containing at least one heterozygous SNV, and discarded those with no heterozygous SNVs. Next, using all BAM files for each individual (11 iPSC BAMs across seven individuals, and 13 iPSC-CM BAMs across seven individuals), we assigned contacts to their matching haplotype when all heterozygous SNVs matched a single haplotype, and discarded other contacts. We did not remap reads with WASP as (1) the alignment scores

from the single end bams do not reflect the true mapping scores of the Hi-C contact due to the highly chimeric nature of Hi-C reads, and (2) the insert size that appears from normal paired-end mapping of Hi-C reads, and thus cannot be filtered by WASP. At each loop, we then calculate a $Z$ score via a binomial approximation to a normal distribution from the greater and lesser allele counts, always using the greater allele as the test variable, and then calculated a $p$-value from a half-normal distribution for each person to account for the imbalance values being > 0.5 by definition. When comparing Hi-C counts across haplotypes, biases known to affect Hi-C read depth are held constant as the genomic loci are held constant (see Methods section "Identification of CTALs" for details). To obtain a single $p$-value for imbalance of each loop, we use Fisher's method to obtain a meta-$p$-value across all seven individuals. Finally, to identify genome-wide significant HTALs, we use the Benjamini-Hochberg FDR correction to obtain a $q$-value, and identified loops with a $q$-value <0.05 as genome-wide significant HTALs.

To identify HTALs with a beta-binomial test, we utilized the combined haplotype scripts from WASP. First, we created a CHT input file using the haplotype counts for each loop. Next, we passed these files to fit_as_coefficients.py to calculate the binomial overdispersion parameters. Finally, we obtained $p$-values for each individual separately from combined_test.py with the option –as_only. These $p$-values were combined via Fisher's method and both the combined and raw $p$-values were used for downstream analyses. This analysis resulted in the identification of 89 HTALs, 83 of which were contained in the half-normal HTAL set (93%). We repeated the analyses from Fig. 6 using these results and observed similar enrichment patterns to the half-normal approach, but found stronger enrichments at imprinted loci (Supplementary Figure 7).

**Calculation of power to detect HTALs**. To determine the power to identify chromatin loop imbalance at different allelic imbalance fractions, we calculated $Z$ scores as above using parameters for numbers of contacts (ranging from 5 to 100 in steps of 5), allelic imbalance fractions (from 0.55 to 0.95 in steps of 0.05). We then calculated the power from a half-normal distribution using alpha thresholds ranging from $1 \times 10^{-x}$ to $9 \times 10^{-x}$ for any integer $2 \le x \le 6$ within each individual. We then calculated the alpha threshold from a meta-$p$-value obtained from combining seven individuals displaying the same imbalance via Fisher's method.

**Chromatin state enrichments at HTALs**. To examine whether any pairs of chromatin states were enriched at opposing HTAL anchors, we annotated all HTAL anchors with the chromatin states they overlapped (with iPSC or fetal-heart chromatin states) via pgltools intersect1D. Next, we used a Fisher's exact test for each pair of states (125 pairs total) to compare the proportion of HTALs with the states at their anchors to the proportion of non-HTALs. Finally, to correct for multiple testing, we performed FDR correction on the $p$-values.

**Loop set enrichments at HTALs**. To examine whether any loop sets (CTALs, cell type called loops, CTCF ChIA-PET interactions, or pHiC interactions) were enriched for HTALs, we annotated HTALs by the loops they overlapped via pgltools intersect. Next, we used a Fisher's exact test for each loop set to test for enrichment of the loop set within HTALs relative to non-HTALs.

**ASE gene and peak identification**. To identify genes and peaks exhibiting genome-wide significant allele-specific expression (ASE) from RNA-seq or ChIP-seq data, within each cell type, for each individual, we pooled all samples by cell type, applied WASP[75] to reduce reference allele mapping bias, used MBASED[46] (R package version 1.4.0) to obtain allelic ratios and $p$-values for each gene and peak for each individual, and identified significant genes or peaks as those with an FDR corrected $q$-value < 0.05.

**Chromatin loop set and genomic feature enrichment for HTALs**. To identify chromatin loops containing imprinted genes or CNVs, we utilized the pgltools findLoops function to create a bed file from the union loop set, and then used bedtools[76] intersect function to obtain all loops containing the genomic characteristic. To identify ASE genes overlapping chromatin loop anchors, we utilized pgltools intersect1D function. To identify eQTLs polymorphic in the family with eGenes connected by a chromatin loop, we created a set of all eQTL-eGene pairs with empirical $p < 0.05$ from DeBoever et al.[32] in the PGL format, and utilized pgltools intersect to find loops within 20 kb of the eQTL-eGene pair. For each genomic feature, we performed a Fisher's exact test across multiple chromatin loop imbalance $p$-value thresholds to determine if the genomic feature was enriched in HTALs over the union loop set. To obtain a $p$-value threshold HTAL set, we filtered all chromatin loops to those exhibiting allelic imbalance with a $p$-value less than or equal to the threshold.

**CNV type analyses**. To measure enrichment of CNV types within union loops and HTALs, we identified all CNVs from DeBoever et al.[32] present in these individuals (1767 deletions and 1045 duplications). We then identified all loops which contained CNVs of the same type using pgltools findLoops and intersect1D. Finally, to obtain $p$-values, we used a binomial approximation to a normal distribution, and tested for an enrichment in duplications above the genome-wide rate ($\mu = 0.37$: the fraction of detected CNVs that were duplications).

**Concordance between loop and molecular phenotype imbalance**. To examine the relationship between molecular phenotype (RNA-seq and H3K27ac ChIP-seq) allelic imbalance and chromatin loop imbalance, we compared allelic differences in molecular phenotype data to chromatin loop imbalance frequencies in iPSC-CM data. We first removed chromatin loops containing imprinted genes or CNVs. Next, for each union chromatin loop, we utilized the aforementioned allelic imbalance data; for each molecular phenotype, we pooled the iPSC-CM reads from all samples for each individual, applied WASP[75] to reduce reference allele mapping bias, and used MBASED to obtain major allele frequencies of each gene/peak. We then identified the most imbalanced SNV in each gene/peak, and used the SNV's phase to determine the maternal allele frequency of the gene/peak. We then converted maternal allele frequencies to fold changes by dividing the maternal allele frequency by the paternal allele frequency for both molecular phenotypes, and the chromatin loop data.

## Data availability

All genomic data are available through dbGAP accessions phs000924 (Hi-C, RNA-seq, CHiP-seq, ATAC-seq) and phs001325 (whole-genome sequence SNV and CNV genotypes). Processed data files are available through GEO entry GSE125540. Data for Figure 2d, e, g, h; Supplementary Figure 2G; 4C, D; 6B, C, D, E; and 7D, E are in the Source Data File; all other figures can be created from the processed data in Supplementary Data files 4–14. Code for correcting switch errors using family structure is available at https://github.com/billgreenwald/HiC-Family-Phaser.

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

## Acknowledgements

This work was supported in part by a California Institute for Regenerative Medicine (CIRM) grant GC1R-06673 and NIH grants HG008118-01, HL107442-05, DK105541-03, and DK112155-01. RNA-seq were performed at the UCSD IGM Genomics Center with support from NIH grant P30CA023100. W.W.G. was supported by the National Heart, Lung, and Blood Institute of the National Institutes of Health under Award Number F31HL142151. D.J. was supported by the National Library of Medicine Training Grants T15LM011271. P.B. was supported in part by the Swiss National Science Foundation Postdoc Mobility fellowships P2LAP3-155105 and P300PA-167612. Whole-genome sequencing was performed at Human Longevity, Inc. Arima Genomics was supported by NIH grant R41HG008118. We would like to acknowledge Yunjiang Qiu for scripts and help with qualitative Hi-C loop calling.

## Author contributions

Conceptualization: W.W.G., H.L., E.N.S., and K.A.F.; methodology: W.W.G., E.N.S., and K.A.F.; software: W.W.G., H.L., and H.M.; validation: H.L.; formal analysis: W.W.G., H.L., P.B., M.D., and D.J.; investigation: W.W.G., H.L., A.D.A., P.B., A.S., and S.S.; data curation: W.W.G. and H.M; writing—original draft: W.W.G., H.L., E.N.S., and K.A.F.; writing—review

& editing: W.W.G., E.N.S., and K.A.F.; visualization: W.W.G. and H.L.; supervision: E.N.S. and K.A.F.; project administration: K.A.F.; funding acquisition: K.A.F.

## Additional information

**Competing interests:** Drs. Anthony Schmitt and Siddarth Selvaraj are employees and stockholders at Arima Genomics. They generated Hi-C libraries and data, and provided advice on phasing and related analyses, but did not influence the scientific outcome of this work. The remaining authors declare no competing interests.

