## [Peer Review File · Nature Communications]

Reviewer #1 (Remarks to the Author):

Greenwald et al. generate Hi-C data for iPSC and iPSC-derived cardiomyocyte across multiple individuals, detect chromatin loops from these data either jointly or in a cell-type and haplotype-specific way, and study their association with chromatin features, eQTLs and gene expression.

While the dataset appears impressive, unfortunately I found it rather difficult to pin down exciting biological insights out of the analyses in their present form. The correlative relationships between chromosomal interactions and the above features have been covered extensively by previous studies in a range of systems, including two recent studies in cardiomyocytes (Choi et al., Nat Comms 2018; Montefiori et al., eLife 2018) that used high-resolution Promoter Capture Hi-C as opposed to conventional Hi-C. Haplotype-specific analyses could provide important additional insights, but the data appears underpowered, and I am not sure I fully agree with the authors' analysis strategy, as well as the final conclusions in this respect. Finally, it appeared rather striking that the authors did not attempt a mechanistic interpretation of their principal findings.

My general recommendation would be that the authors (a) streamline and compress the presentation of known and/or expected correlative findings, (b) brush up the analysis of haplotype-resolved data and clearly highlight its limitations, and (c) clearly present their interpretation of the nature of the detected loops and their differential strength. Specific points to this extent and beyond are listed below.

Major points:

1. How do the authors conceptualise the detected loops as compared with, for example, interactions between TAD boundaries or enhancer-promoter interactions? At a technical level, do they believe promoter-based chromatin loops to represent just a small subset of a much larger number of such contacts typically detected with HiChIP or Capture Hi-C, with the rest of them undetectable in their analysis due to the lower coverage? If they believe chromatin loops to possess some distinct biological properties as compared with the above-mentioned types of contact, they need to provide an interpretation of their findings in the context of these properties specifically as opposed to just chromatin contacts in general.

2. Likewise, what is the authors' mechanistic interpretation of loop strength, and "small quantitative differences" therein within and across haplotypes? A contact in a given cell at a given moment of time is a binary property - it's either there or not. Quantitative differences are a population-level phenomenon.

3. It is hard to argue with the authors' observation that haplotype-resolved analysis did not identify significant allelic differences in loops. However, I wonder to what extent this can be due to (a) the

insufficient statistical power of the dataset and (b) distinct mechanistic properties of the kind of chromosomal contacts they recover?

4. It is almost certain that the binomial test is not conservative enough for detection of allelic differences. I wonder therefore why the authors only used WASP for local realignment and not for beta-binomial-based ASE testing that seems fit for purpose and was developed specifically because allele-specific read counts are overdispersed. Is this because it pretty much found nothing? If this is the case, I would recommend restricting the analysis fully to data pooled across multiple loops that share similar properties.

5. The authors interpret the weak (in terms of effect size), but statistically significant association between the molecular phenotypes and loop strength as an indication that “small changes in chromatin looping are likely functionally relevant”. However, it’s unclear why they believe them to be causal for these changes rather than reflect the subtle (and potentially artifactual) effects of chromatin opening (associated with H3K27ac) and transcription on Hi-C contact frequency.

Minor points:

Bottom of p.2. Sentence starting with “These contradictory results...” is way too long and barely legible.

Last paragraph of Introduction (p.3). Unclear what is meant by “these analyses did not identify particular functional loops”.

p.4. The authors state: “We therefore pooled the data within each cell type to create the highest resolution chromatin maps (~2kb map resolution) in iPSCs and iPSCCMs”. However, resolution is the property of the analysis rather than data, which is defined by coverage. The authors should therefore either present this statement in terms of coverage or at least clearly indicate how they tuned the resolution based on the latter.

p.4. Comparison with the number of loops in the Rao data is only meaningful if exactly the same algorithm was used for their detection. However, it’s unclear if it indeed was the case.

p.4. “Additionally, we examined the directionality of CTCF motifs using meme...” I would strongly suggest clearly mentioning in the text and not just in the figure that only a minority of loops actually contained detectable CTCF motifs before proceeding to describe their orientation.

p.4. The authors don't seem to provide any interpretation to the potentially interesting observation of active and inactive loop networks "crossed over through the bivalent states in iPSCs". Does this implicate interactions between active and bivalent loci? Would be good to discuss this in the context of Freire-Pritchett et al., eLife 2017 and Mas et al., Nat Genet 2018.

Bottom of p.4. Do the authors specifically mean to say that Polycomb-mediated loops are larger and more complex than active loops or do they mean any non-active loops overall? Would be good to clarify.

p.5. It's confusing to mention edgeR and then proceed to describe log-CPM-based analysis. I would only mention it further down, when raw count-based analysis is introduced.

p.5. TMMs need disambiguation in the Results section.

p.5. What is the authors' interpretation of the fact that only iPSC-CTAL, but not CM-CTAL anchors are enriched for differential active enhancers?

p.6. "We first examined whether loops which colocalize iPSC-eQTLs (previously identified from a cohort including these individuals³²) to the genes that they were statistically associated with (eGenes) had stronger contact intensities within iPSCs than iPSC-CMs. We found a strong enrichment (Mann Whitney-U $p \sim 1 \times 10^{-293}$) for iPSC contact intensity above non eQTL-eGene loops (Figure 4A), indicating that loops with higher strength in a cell type may be more likely to harbor functional genetic variation".

It seems that the first sentence introduces a different kind of analysis to the one presented in the second one: the first one mentions tissue-specificity, while the second one seems to suggest comparison with non eQTL-eGene loops within a single cell type. This may just be a matter of wording.

p.9. Unclear sentence: "For these analyses, while data is pooled across loci to determine a genome wide trend, each pair of fold changes of phenotype and loop strength are not affected by the aforementioned biases to Hi-C read depth, as these are held constant when examining the same set of loop anchors across haplotypes".

Reviewer #2 (Remarks to the Author):

Decision:

Major revision.

Summary:

The authors seek to address a fundamental question regarding the 3D genome: do DNA mutations affect the 3D genome and to which extent? Until now, only a limited number of studies showed a link between mutations such as SNPs and 3D genome: Rao et al. (Cell 2014) using 1kb resolution Hi-C data from one cell line, Tang et al. (Cell 2015) using CTCF/RNA POL II ChIA-PET data from 3 patients and Mumbach et al. (Nature 2017) using H3K27ac/CTCF HiChIP on some cell lines. Others showed the role of structural variants on 3D genome (Lupianez et al., Cell 2016).

Here, the authors generated a resource of phased Hi-C contact maps in induced pluripotent stem cells (iPSCs) and in iPSC-derived cardiomyocytes (iPSC-CMs) from 7 patients, for which genome sequence (50X WGS), gene expression (RNA-seq) and enhancer activity (H3K27ac ChIP-seq) data were also generated (Benaglio et al., BioRxiv 2018). The authors then use allelic imbalance to assess the effect of SNPs, CNVs and imprinting on chromatin looping, as well as expression and enhancer activity.

Among the contributions, the authors show quantitative differences in loop strength across cell types, instead of loop presence/absence depending on cell type. The authors also found that small changes in loop strength is associated with large changes in gene expression and enhancer activity. Finally, the study revealed that a strong impact of CNVs and imprinting on 3D genome, unlike SNPs.

Overall, the study provides novel insights into the genetic determinants of the 3D genome and its link with gene expression and enhancer activity, as well as a useful 3D genome data resource for further investigations. Moreover, the article is clear and well written.

Major revision:

The Hi-C allelic imbalance results presented here are weak due to the improper use of statistical test to rigorously assess allelic imbalance. Since Hi-C allelic imbalance results represent the core of this study, this is an important issue. To assess Hi-C allelic imbalance, the authors "calculate a Z score via a binomial approximation to a normal distribution" (1st paragraph, p29). There are two main problems with this very basic test. First, this is an approximation and thus the p-value might not be accurate when the count are low. This is problematic here because the Hi-C data are typically sparse and the counts are often low. Note that the binomial exact test should be used instead of the normal approximation test when no overdispersion is found in the data. Second, NGS counts are known to be overdispersed and thus do not follow a binomial law. Not accounting for overdispersion might represent a big issue, since it might lead to many false positive variants. To account for overdispersion, different tests were proposed that overcome the binomial exact test and account for overdispersion, that are implemented for instance in WASP [1] and MBASED [2] or any other dedicated program. The authors should use a program dedicated to assess allelic imbalance that will account for overdispersion and biases.

It is likely that the authors will find a smaller number of significant Hi-C allelic imbalances (HTALs) using a dedicated program than detected using the normal approximation (114 HTALs). If the authors detect only a few number of significant HTALs using a dedicated program which therefore do not allow them to integrate with other data, and thus prevent further key analyses in the article, I believe the authors should keep the normal approximation test they used but they should clearly state in the article the weakness of the test they used (and thus many false positives might be detected), explain why they used it instead of a test accounting for overdispersion and that the results should be interpreted with caution. Another option might be to take an FDR=10% or 20% and keep a lower number of loops for testing (only the strongest loops above a certain threshold) to reduce p-value correction effect.

Ref:

[1] van de Geijn, B., McVicker, G., Gilad, Y. & Pritchard, J. K. WASP: allele-specific software for robust molecular quantitative trait locus discovery. *Nat Methods* 12, 1061- 1063, doi:10.1038/nmeth.3582 (2015).

[2] Mayba, O. et al. MBASED: allele-specific expression detection in cancer tissues and cell lines. *Genome Biol* 15, 405, doi:10.1186/s13059-014-0405-3 (2014).

Reviewer #3 (Remarks to the Author):

This paper describes interesting findings about the role of changes in quantitative loop strength in biological outcome. This is important work, as it challenges the assumption that only complete loss or gain of loops will be biologically meaningful, and it complements previous observations that changes in major architectural factors have sometimes surprisingly subtle effects on genome organization. However, given that the study places extensive emphasis on subtle and quantitative changes in loops, I am concerned about some of the steps taken to define loops. Admittedly, loop calling is a hard problem with no one right answer. But I am concerning that they tuned their loop-calling algorithms to focus on the strongest “dots” in the Hi-C heatmap, presuming that these are the most biologically functional and gene regulatory loops, when these may be the most invariant, primarily architectural features. Also concerning is the pooling of individuals or cell types, which may blur out specific loops that would have been observed in specific samples. There are also places in the haplotype section where the writing of the manuscript becomes unclear and hard to follow, and could use revision.

My detailed comments are below:

The authors state “As chromatin structure has been shown to be an evolutionarily stable trait, changes in chromatin loops across cell types are a priori more likely to take the form of subtle strength modulations than the formation of entirely new loops” This is an interesting point, and it is indeed worth noting that changes may be subtle, but this ignores the different levels of variation at different scales of chromatin structure: A/B compartments change notably across cell types, and small scale chromatin structure (local compaction and accessibility) likewise varies, as may local enhancer promoter connections. It is mostly the CTCF-mediated TAD-scale loops that are stable across syntenic regions and cell types. The authors should be careful about their definitions and language: if they mean certain types of loops, then they should not imply this encompasses all chromosome structure.

I am concerned about the justification for pooling Hi-C data from different individuals for the same cell type. The Pearson correlations are calculated between samples with 1 Mb bins. First, it has been shown that Pearson correlation is not a good way to evaluate Hi-C similarity (Yaffe and Tanay, 2011; HiCRep tool, Yang et al, 2017; Yardimci, BioRxiv, 2017). Second, even if Hi-C datasets are similar at 1 Mb resolution, there is no guarantee that they are similar at loop-scale (10-500 kb resolution). I understand that high numbers of reads are needed for loop calling, so pooling more reads could increase resolution and power, but if the whole argument is to look for subtle changes in loop strength, isn't it quite likely that some subtle changes will be averaged out across individuals? I suggest that the authors more deeply sequence 1 individual of each cell type and check how the called loops of that single individual compares between cell types and to the pool of the other individuals.

Calling loops at multiple resolutions and with multiple algorithms is a great idea, but I am not sure it is justifiable to use “visual best match” as the criteria for whether loop calling is working well. It is an oversimplification to state that “Fit-Hi-C is known to detect false positive loops along TAD boundaries” Why should these be called false positives? These lines along boundaries are likely due to loop extrusion and represent real enrichment of interactions between one boundary (loop anchor) and other regions that slide past. These may be biologically meaningful interactions. On the flip side, we don't know that the prominently visible “dots” in Hi-C maps (assumed here to be “true positives”) are actually more meaningful. Indeed, these TAD corner peaks may be the most invariant and thus least likely to relate to changes in gene expression. So, I'm concerned that the authors may have artificially tuned their algorithms to detect visually obvious peaks while unjustifiably excluding others. If this kind of “visually correct” filtering is not done, would there be more correlation with gene expression differences and called peak changes?

The idea of using raw reads from all replicates at each loop to define differences between cell types is nice, and it is a key observation that loops not “called” in one cell type often still have similarly high contacts around that region. But I'm concerned that by looking at raw reads just at the loop location (and nothing surrounding it), a presumed difference in “loop strength” could in fact be a

difference in overall interaction decay with distance between the samples (due to an overall A/B compartment or compaction change, for example). Relatedly, Figure 2A shows a very nice example of cell-type specific differences, but reducing this to the few called loops seems incomplete: iPSCs have a whole domain structure that is not visible in CM cells.

What about checking the profile of iPSC H3K27Ac or CTCF at CM-specific peaks? Is there indeed less enrichment?

Page 5: There are references to figure 2G, but this figure doesn't exist. Should this be 3G? Either way, I can't see how the statements made are supported by the figure (where are the regions enriched for heart and iPSC bivalent chromatin? That can't just be the generic category 12_EnhBiv...

It would be well for the authors to consider, and adjust their language accordingly, what is actually meant by "loop strength". This is likely a measure of % of the cell population that had the loop, which probably should at least once be clarified to mean "loop frequency". The number of contacts does not tend to directly correspond to "the distance in 3D space between two anchors" as claimed on page 7.

It is worth noting that previous work has suggested that the degree of difference between haplotypes may vary across cell types (Rivera-Mulia, 2018 shows a good example of this in replication timing: ES cells have more variation between haplotypes than differentiated cell types) So, this makes it a little dangerous to pool Hi-C data between iPSC and CM for haplotype comparison. Also, it then seems a circular argument to pool cell types to call HTAL, but then claim to find that haplotype variable loops are not cell type variable. (Didn't the pooling of cell types make it much harder to identify haplotype variable loops that also vary between cell types?)

The last two sentences on page 8 are very confusing, possibly due to errors: "this observation is consistent with linearly closer loci having increased Hi-C contact strength, and thus more power for detection, as deletions reduce the linear space between loci increasing contact strength thereby increasing power, and duplications increasing the space increasing contact strength thereby decreasing power. Overall, these results confirm that genetic imprinting, and suggest that CNVs, may be strong drivers of allelic imbalanced chromatin looping."

How can both reducing and increasing the space increase contact strength? What do you mean "results confirm that genetic imprinting"? Is it fair to infer that raw contacts indicate loop strength among regions whose linear distance has been changed? Don't you need to compare the loop contacts to surrounding contacts to see if the relative loop strength has actually changed?

Minor points:

Fig S1A refers to “regular read pairs” and “normal split read pairs”. It is not clear what “regular” means here, nor is that defined on the Juicer output or documentation, so this should be clarified.

Fig S1C is a nice graph, but I’m very surprised that nearly 100% of all 5 kb bins would have 2000 contacts per bin, even in the interchromosomal interaction space. Does this set of bins truly include all contact bins in the genome? Or is it the sum of contacts involving every given bin?

Fig S1 legend has a typo: “edges connecting statistically significant pairs of chromatin states found at opposing.” (opposing what?)

Make sure Fig S4C is a high resolution enough figure for readers to see the pink and blue squares you’re drawing (I can’t make them out)

Page 4: MEME should be capitalized

Page 7: “ASE genes/peaks” are not defined (what does “ASE” stand for?) and make subsequent paragraphs hard to read

We would like to thank the three reviewers for their thoughtful feedback and helpful suggestions on how to improve our manuscript. For each specific concern, we have responded below with the changes that we made. Overall, we have re-written the introduction and discussion in order to reframe the biological insights derived from our study, as suggested by multiple reviewers. We have also transitioned away from using the term loop strength, and instead use contact propensity, and have integrated literature supporting how contact frequency is a measure of contact propensity and spatial colocalization. Finally, to aid in clarity, we have made a number of edits to the text and the title. We believe that these changes have greatly improved the clarity, relevance, and strength of the observations in our manuscript.

Reviewer #1 (Remarks to the Author):

Greenwald et al. generate Hi-C data for iPSC and iPSC-derived cardiomyocyte across multiple individuals, detect chromatin loops from these data either jointly or in a cell-type and haplotype-specific way, and study their association with chromatin features, eQTLs and gene expression.

While the dataset appears impressive, unfortunately I found it rather difficult to pin down exciting biological insights out of the analyses in their present form. The correlative relationships between chromosomal interactions and the above features have been covered extensively by previous studies in a range of systems, including two recent studies in cardiomyocytes (Choi et al., Nat Comms 2018; Montefiori et al., eLife 2018) that used high-resolution Promoter Capture Hi-C as opposed to conventional Hi-C. Haplotype-specific analyses could provide important additional insights, but the data appears underpowered, and I am not sure I fully agree with the authors' analysis strategy, as well as the final conclusions in this respect. Finally, it appeared rather striking that the authors did not attempt a mechanistic interpretation of their principal findings.

My general recommendation would be that the authors (a) streamline and compress the presentation of known and/or expected correlative findings, (b) brush up the analysis of haplotype-resolved data and clearly highlight its limitations, and (c) clearly present their interpretation of the nature of the detected loops and their differential strength. Specific points to this extent and beyond are listed below.

We would like to thank the reviewer for their helpful feedback and have addressed most of these concerns as specific points below. We also appreciate the reviewer pointing out the promoter Hi-C articles and we have now included them in the manuscript. We agree that it has been recently established that loops tend to overlap functional regions (i.e. eQTL, chip-seq peaks, and GWAS signals), and that more loops tend to indicate higher expression (shown by these PHiC papers Choy et al and Montefiori et al, as well as other HiC papers such as Bonev et. al). We believe that our study extends these analyses in two ways: 1) our analyses

examine the difference in contact frequency at the loops, and how these quantitative differences are associated with differences in gene expression and regulation, rather than asking if a loop is present or absent, and 2) our analyses examine many types of chromatin loops, rather than only promoter-based loops. These analyses show that that contact propensity at loops is a continuum which is tied to changes in gene expression and regulation at a continuous and quantitative level. Additionally, we find a quantitative relationship between contact propensity and H3K27ac and observe this across all kinds of loops; not just promoter enhancer loops. These other loops are missed by pHiC. Thus, we feel that our work extends that of these recently published papers.

To clarify the novelty of our manuscript, we have substantially changed the Intro, Results, and Discussion as follows:

Introduction (pg3): Chromatin loops colocalize regulatory elements with their targets¹⁻¹⁵ by bringing genomic regions that are distant from one another in primary structure close together in 3D space¹⁶. These colocalized regions, also known as loop anchors, are preferentially enriched for disease associated distal regulatory variation and expression quantitative trait loci (eQTLs)¹⁷⁻²². While it has been shown that the physical 3D distance between looped loci can vary^{16,23-25}, previous studies examining cell type and haplotype differences in looping have considered loops to be either present or absent, rather than as a quantitative phenotype. Thus, the extent to which quantitative differences between chromatin loops exist, and whether they are associated with differences in gene expression and regulation, has yet to be explored. Moreover, the extent that regulatory genetic variation acts by affecting the physical distance between looped loci is not yet known.

Introduction (pg3): Bulk chromatin conformation assays (e.g. 3C, 4C, and Hi-C) were designed to identify genomic regions in close spatial proximity¹⁶ by measuring physical contact frequency between two pieces of colocalized (ie looped) DNA in a pool of cells. As an individual contact is a physical interaction in a single cell, the contact frequency measured in a pool of cells reflects the probability for the interaction to occur (**contact propensity**) across all cells in the sample. Overall contact frequency is a function of spatial proximity, which has enabled Hi-C data generated in either bulk²⁶⁻²⁹ or single cell³⁰ assays to be used to reconstruct the physical 3D structure of chromatin at the compartment, TAD, and loop levels by incorporating polymer physics models. Additionally, single cell Hi-C has shown that while single cell contacts occur at loops called from bulk data, individual cells vary in the predicted physical distances between their looped loci³⁰. Moreover, fluorescence in situ hybridization studies have shown that spatial distance and contact frequency between looped loci vary concordantly²³⁻²⁵. Together, these studies imply that differences in contact propensity reflect differences in the physical distance between the looped loci. Therefore, investigating contact frequency as measured by Hi-C, in combination with molecular phenotypes, may reveal if the physical distance between looped loci varies across cell types, and if this is associated with differential regulation of gene expression.

Results (pgs 5-6): In both cell types, we found enrichments for active and bivalent chromatin states (Figures 2C & S1D), H3K27ac (Figure 2D left & S1E), and chromatin accessibility (Figure 2D middle & S1F) from their respective cell type above shuffled null loop sets. Additionally, we found that 45.5% of loops had CTCF motifs at both anchors, and that across all loops, CTCF motifs were centrally enriched at anchors (Figure 2D right). As seen in Rao et. al², the vast

majority of loops (85.3%) with CTCF motifs at both anchors had inward facing CTCF motifs. Further, 63.3% and 65.3% loops in iPSC and iPSC-CMs, respectively, were within 25kb of a CTCF ChIA-PET interaction from GM12878⁵. We next examined the types of chromatin states that were statistically significantly paired together (Fisher's Exact $p < 0.05$) and found two subnetworks, one with active chromatin states and the other with repressed or bivalent chromatin, which were discrete in iPSC-CMs (Figure 2E) and crossed over through the bivalent states in iPSCs (Figure S1H). This crossover, which is only present in iPSCs, is consistent with the role of bivalent and polycomb chromatin in pluripotency⁴⁰⁻⁴², the role of bivalency in maintaining stem cell region connectivity⁴², and with the shift of active states to bivalent and polycomb during differentiation and chromatin rewiring⁴³. This result suggests that these specialized roles of bivalent and polycomb chromatin extend to the fine-scale aspects of chromatin architecture, including loops. We next examined the consistency of these loops with previously identified promoter loops from promoter capture Hi-C (**pHiC**) and found 28.7% and 33.5% of iPSC and iPSC-CM loops to be within 25kb of a pHiC interaction in these cell types, respectively.

Discussion (pg12): We performed quantitative comparisons of contact frequency across cell types and haplotypes to identify differences in chromatin looping, and integrated these differences with quantitative measures of differential expression and H3K27ac to examine the functionality of contact propensity. These analyses revealed a proportional association between contact frequency and gene expression/H3K27ac, which surprisingly linked the phenotypes across different magnitudes of variability: extremely subtle changes in contact frequency were associated with large differences in gene expression and H3K27ac. If contact propensity at loops is a fundamental regulator of gene expression, differences in contact propensity would be expected to be associated with similarly sized differences in gene expression regardless of the environment in which the differences occurred (ie across cell type, haplotype, or experimental conditions). As we observed a consistent relationship between the two, we believe these data indicate that contact propensity is a mechanism involved in regulating gene expression, similar to enhancer activity or transcription factor binding strength. 3D modeling of the genome from Hi-C^{26-30,52}, and fluorescence in situ hybridization experiments^{23,25,53}, have shown a relationship between contact frequency and spatial proximity, which suggests that loci with increased spatial proximity (ie are closer together) come into physical contact more easily, and thus produce a Hi-C contact more readily. Extending our results to spatial proximity, our data suggests that genetic variation at chromatin loops may exert their effects on gene expression or gene regulation by modulating the spatial proximity between the loci of, and thus contact propensity at, chromatin loops. Notably, as we identified a non-directional correlation, contact propensity may either affect, or be affected by, gene expression and/or regulation. In the latter case, genetic variation would mediate effects on spatial proximity through affecting gene expression or regulation.

Major points:

1. How do the authors conceptualise the detected loops as compared with, for example, interactions between TAD boundaries or enhancer-promoter interactions? At a technical level, do they believe promoter-based chromatin loops to represent just a small subset of a much larger number of such contacts typically detected with HiChIP or Capture Hi-C,

with the rest of them undetectable in their analysis due to the lower coverage? If they believe chromatin loops to possess some distinct biological properties as compared with the above-mentioned types of contact, they need to provide an interpretation of their findings in the context of these properties specifically as opposed to just chromatin contacts in general.

As the reviewer suggests, we believe (in line with Rao et. al 2014 and Figure 3B) that the loops identified with Hi-C represent multiple types of loops including promoter loops identified with promoter capture Hi-C, structural loops such as those identified with CTCF ChIA-PET, as well as others (e.g. polycomb complexes or enhancer centric loops). We thus called TADs using an HMM approach, and compared loop calls to pairs of TAD boundaries. We found the majority of the loops to not overlap TAD boundaries (~95%). To make these distinctions clearer in the manuscript and explore whether different subtypes of loops show different associations, we have now performed a direct comparison between our loops and promoter capture Hi-C or CTCF ChIA-PET loops and report our findings in the Results (see below). We obtained CTCF ChIA-PET from GM12878 (Rao et. al 2014), and promoter capture Hi-C from iPSCs and iPSC-CMs (Montefiori et. al, 2018), examined the overlap in these data with our loops. These analyses confirm that our loop sets have high overlaps with promoter centric and structural loops, in addition to the other types of loops that we highlighted in Figure 2F. However, as expected, we do not identify all loops from the capture Hi-C methods, as capture highly enriches the sequenced library and thus adds power for identifying a specific type of loop. We have also updated the Results as follows to make these points clearer:

Results (pg5): To examine whether these loops were predominantly demarcating TADs, or were separate from TAD structure, we also called TADs in both cell types and examined the number of loops that had both anchors within 25kb of TAD boundaries. We found only 2.9% of iPSC loops, and 5.1% of iPSC-CM loops, to have both anchors at TAD boundaries, indicating that these loops were primarily not demarcating TADs. These iPSC and iPSC-CM called loop sets provide a resource for the analysis of long range gene regulation across the genome.

Results (pgs 5-6): Additionally, we found that 45.5% of loops had CTCF motifs at both anchors, and that across all loops, CTCF motifs were centrally enriched at anchors (Figure 2D right). As seen in Rao et. al², the vast majority of loops (85.3%) with CTCF motifs at both anchors had inward facing CTCF motifs. Further, 63.3% and 65.3% loops in iPSC and iPSC-CMs, respectively, were within 25kb of a CTCF ChIA-PET interaction from GM12878⁵. We next examined the types of chromatin states that were statistically significantly paired together (Fisher's Exact $p < 0.05$) and found two subnetworks, one with active chromatin states and the other with repressed or bivalent chromatin, which were discrete in iPSC-CMs (Figure 2E) and crossed over through the bivalent states in iPSCs (Figure S1H). This crossover, which is only present in iPSCs, is consistent with the role of bivalent and polycomb chromatin in pluripotency⁴⁰⁻⁴², the role of bivalency in maintaining stem cell region connectivity⁴², and with the shift of active states to bivalent and polycomb during differentiation and chromatin rewiring⁴³. This result suggests that these specialized roles of bivalent and polycomb chromatin extend to the fine-scale aspects of chromatin architecture, including loops. We next examined the consistency of these loops with previously identified promoter loops from promoter capture Hi-C (**pHiC**) and found 28.7% and

33.5% of iPSC and iPSC-CM loops to be within 25kb of a pHiC interaction in these cell types, respectively. Together, these results indicate that the identified chromatin loops include those with active regulatory interactions (e.g. promoter-enhancer interactions), those with repressive interactions (e.g. polycomb complexes), structural loops (CTCF-CTCF), and those with a variety of other types of chromatin states (that were not significantly enriched for being paired together) at their anchors.

2. Likewise, what is the authors' mechanistic interpretation of loop strength, and "small quantitative differences" therein within and across haplotypes? A contact in a given cell at a given moment of time is a binary property - it's either there or not. Quantitative differences are a population-level phenomenon.

We agree that our rationale for studying loop strength/contact intensity was not clear. We have now rewritten the Introduction and included new next in the Discussion that we hope clarifies our approach and the implications of our study. These changes can be found above in our response to the reviewer's summary statement.

3. It is hard to argue with the authors' observation that haplotype-resolved analysis did not identify significant allelic differences in loops. However, I wonder to what extent this can be due to (a) the insufficient statistical power of the dataset and (b) distinct mechanistic properties of the kind of chromosomal contacts they recover?

a) We believe that our findings suggest that loop imbalance associated with regulatory changes are quite small and therefore we agree that the reason that we did not observe HTALs associated with ASE or gene regulation was due to our power to detect HTALs with > 70% imbalance. In order to better emphasize and clarify this point, we have made the following changes in the results and discussion:

Results (pg 10): These results and power indicate that while we may not detect all small haplotype differences (ie those with imbalance < 70%), large haplotype differences in chromatin looping occur infrequently

Results (pg 9): We observed a median of 50 informative contacts per individual per loop, which corresponds to 100% power to identify HTALs with an allelic imbalance ratio of 70% or higher with $\alpha = 0.02$ in an individual (Figure S5C), or at $\alpha = 2 \times 10^{-5}$ when all samples display similar imbalance and are combined with Fisher's method meta-analysis

Discussion (pg 12): As the effects of imprinting are parental in nature, rather than genetic, it is necessary to search outside of these regions for causal regulatory variants. In non-imprinted regions, if we interpolate the association between gene expression and contact propensity, the linear model would suggest that a gene with 98% ASE would be expected to be associated with a loop imbalance of only ~52%. This minute difference in loop imbalance provides a possible explanation for why we did not observe HTALs associated with gene regulation or ASE, but found a quantitative association between Hi-C signal and functional phenotypes overall.

b) While we had previously included an analysis of whether HTALs were more likely to be promoter-enhancer loops (Figure 6E), we agree that a more comprehensive analysis of loop classes would improve the study. We have now performed a pairwise enrichment of chromatin states at HTALs compared to the union loop sets, and observed that no pairs of chromatin states were statistically enriched or depleted for being HTALs (FDR $q > 0.05$). Additionally, we examined whether loops that overlapped established pHiC interactions or CTCF-CTCF ChIA-pet interactions were enriched in HTALs, and found none to be enriched at a nominal p-value ($p > 0.05$). These results suggest that HTAL differences are not confined to specific sub-types of loops. We have now updated the Results with this analysis:

Results (pg 10): We compared the proportion of the 114 HTALs that were also iPSC-CTALs, CM-CTALs, iPSC called, or iPSC-CM called loops to the corresponding proportion of union loops. However, we found no significant differences for any association ($p > 0.05$ for all tests; Figure 6A). We next examined whether a particular type of loop was enriched within HTALs (ie CTCF loops, promoter-enhancer loops; **see Methods**), and found no significant enrichment (FDR $q > 0.05$); together, these results indicate that loops which varied between haplotypes were not more likely to be a specific type of loop.

Methods (pg 32): **Chromatin state enrichments at HTALs**

To examine whether any pairs of chromatin states were enriched at opposing HTAL anchors, we annotated all HTAL anchors with the chromatin states they overlapped (with iPSC or fetal-heart chromatin states) via pgltools intersect1D. Next, we used a Fisher's Exact test for each pair of states (125 pairs total) to compare the proportion of HTALs with the states at their anchors to the proportion of non-HTALs. Finally, we FDR corrected the p-values for multiple-testing control.

Loop set enrichments at HTALs

To examine whether any loop sets (CTALs, cell type called loops, CTCF ChIA-PET interactions, or pHiC interactions) were enriched for HTALs, we annotated HTALs by the loops they overlapped via pgltools intersect. Next, we used a Fisher's Exact test for each loop set.

4. It is almost certain that the binomial test is not conservative enough for detection of allelic differences. I wonder therefore why the authors only used WASP for local realignment and not for beta-binomial-based ASE testing that seems fit for purpose and was developed specifically because allele-specific read counts are overdispersed. Is this because it pretty much found nothing? If this is the case, I would recommend restricting the analysis fully to data pooled across multiple loops that share similar properties.

We agree that the binomial test could be affected by overdispersion in the read counts and have now explored the extent that it affected our results. We utilized the WASP package to estimate overdispersion in Hi-C reads and calculate ASE per individual with a beta-binomial test. After meta-analyzing these p-values with Fisher's Method, we identified 89 HTALs, rather than 114. These 89 HTALs were enriched for imprinted loci at an even higher rate than the 114 loops identified from the half-normal distribution and showed overall similar results for the

analyses that we performed in Figure 6 for HTALs. As the results and interpretations were broadly equivalent, we chose to keep the half-normal approach in the main text, but have now added descriptions of these new analyses in the Results, Methods, and as a Supplemental Figure.

Results (pg10): Next, for each individual, we identified imbalance via a Z score using a half normal distribution (see Methods for details and for complementary analysis using WASP), following which we combined the p-values across individuals with Fisher's method for meta-analysis.

Methods (pg31): To identify HTALs with a beta-binomial test, we utilized the combined haplotype scripts from WASP. First, we created a CHT input file using the haplotype counts for each loop. Next, we passed these files to `fit_as_coefficients.py` to calculate the binomial overdispersion parameters. Finally, we obtained p-values for each individual separately from `combined_test.py` with the option `-as_only`. These p-values were combined via Fisher's method and both the combined and raw p-values were used for downstream analyses. We repeated the analyses from Figure 6 using these results and observed similar enrichment patterns to the half-normal approach (Figure S7).

Additionally, we would like to apologize for the confusion surrounding the usage of WASP. While we used the WASP package to remap the reads from the RNA-seq and H3K27ac ChIP data, we did not apply to the Hi-C reads. Our rationale for this was that a large number of Hi-C reads do not align properly as they are "split" or "chimeric and are also not considered paired nature during alignment (the paired status is re-introduced during QC and filtering with the Juicer pipeline). The WASP pipeline chooses to exclude reads by determining whether a read aligns to a new location better when the reference allele is changed to the alternate allele by examining alignment score – however, the chimeric nature of the reads makes the alignment scores volatile and unreflective of the true strength of the alignment. Additionally, the paired end information cannot be taken advantage of without the Juicer pipeline. Thus, for Hi-C data, WASP would filter out a large number of reads that should be kept. Indeed, we were unable to identify a manuscript that used WASP to remap Hi-C reads. While individually the lack of remapping by WASP may result in some mismapped reads, we expect that the overall variation would be captured by the overdispersion parameters estimated via WASP for the beta-binomial test. As the beta-binomial resulted in the same trends and results, we therefore do not believe that our data is heavily biased by these effects. We have clarified this in the methods of the manuscript as follows:

Methods (pg31): We did not remap reads with WASP as the alignment scores from the single end bam do not reflect the true mapping scores of the Hi-C contact due to the highly chimeric nature of Hi-C reads, and the "insert size" that appears from normal paired-end mapping of HiC reads, and thus cannot be filtered by WASP.

5. *The authors interpret the weak (in terms of effect size), but statistically significant association between the molecular phenotypes and loop strength as an indication that “small changes in chromatin looping are likely functionally relevant”. However, it’s unclear why they believe them to be causal for these changes rather than reflect the subtle (and potentially artifactual) effects of chromatin opening (associated with H3K27ac) and transcription on Hi-C contact frequency.*

We agree that the association we observe does not prove causality and did not intend to imply that we identified a causal relationship. We have now removed the sentence in the Results, and clarified in the Discussion that our findings indicate that either contact propensity affects gene expression, or that gene expression affects chromatin loop strength (and the analog for H3K27ac and loop strength).

Discussion (pg12): Notably, as we identified a non-directional correlation, contact propensity may either affect, or be affected by, gene expression and/or regulation. In the latter case, genetic variation would mediate effects on spatial proximity through affecting gene expression or regulation.

Minor points:

Bottom of p.2. Sentence starting with “These contradictory results...” is way too long and barely legible.

We have split the sentence into multiple sentences and separated out the ideas. We hope this makes it clearer:

Intro (pg3): These contradictory results are likely due to the experimental design and types of effects examined in these studies. Rao et al. used Hi-C data to look for large differences across haplotypes, and thus may have missed smaller effects. The studies using ChIA-PET and Hi-ChIP sought to identify allelic imbalance of all sizes, but employed experimental approaches that may be biased as they simultaneously measure either CTCF binding or regulatory region activity and chromatin looping, thereby conflating the allelic bias of the two phenotypes.

Last paragraph of Introduction (p.3). Unclear what is meant by “these analyses did not identify particular functional loops”.

We agree that this statement was confusing. As we originally added the statement to alleviate confusion about our correlation based analysis, we have removed it from the manuscript.

p.4. The authors state: “We therefore pooled the data within each cell type to create the highest resolution chromatin maps (~2kb map resolution) in iPSCs and iPSCCMs”. However, resolution is the property of the analysis rather than data, which is defined by coverage. The authors should therefore either present this statement in terms of coverage or at least clearly indicate how they tuned the resolution based on the latter.

We apologize for the confusion. The map resolution metric that we are referring to was described in Rao et al, the first study to examine *in situ* Hi-C data and create the high resolution GM12878 map, and is a metric based on the bin size for genomic loci. It is defined as the bin size where at least 80% of loci at that size contain 1000 or more contacts in 1-D (i.e. one anchor of a contact lies in the bin). We have now clarified this use of the term in the Results:

Results (pg. 5): We pooled the data across samples for each cell type, resulting in reference chromatin maps with the highest resolution (~2kb matrix resolution, defined as the resolution at which 80% of loci have 1000 or more contacts with any other locus²) in iPSCs and iPSC-CMs (or any other iPSC derived cell type) to date and were comparable in resolution to the Hi-C map in GM12878² (Figures 1C & S1C).

p.4. Comparison with the number of loops in the Rao data is only meaningful if exactly the same algorithm was used for their detection. However, it's unclear if it indeed was the case.

We agree that as the loop caller will affect the number of loops identified, this is an unnecessary statement. We have removed it.

p.4. "Additionally, we examined the directionality of CTCF motifs using meme..." I would strongly suggest clearly mentioning in the text and not just in the figure that only a minority of loops actually contained detectable CTCF motifs before proceeding to describe their orientation.

We have clarified the number of union loops containing two or more CTCF motifs in the results as follows:

Results (pg 5): Additionally, we found that 45.5% of loops had CTCF motifs at both anchors, and that across all loops, CTCF motifs were centrally enriched at anchors (Figure 2D right). As seen in Rao et. al², the vast majority of loops (85.3%) with CTCF motifs at both anchors had inward facing CTCF motifs.

p.4. The authors don't seem to provide any interpretation to the potentially interesting observation of active and inactive loop networks "crossed over through the bivalent states in iPSCs". Does this implicate interactions between active and bivalent loci? Would be good to discuss this in the context of Freire-Pritchett et al., eLife 2017 and Mas et al., Nat Genet 2018.

We thank the reviewer for suggesting these papers agree that interpreting this observation in the context of this work improves the relevance of these findings. We believe that the interactions we observe between active and bivalent loci are consistent with polycomb chromatin maintaining stem cell regions connectivity (Mas et. al), and that the loss of this crossover in the iPSC-CM is consistent with a shift from active to bivalent and polycomb states during differentiation (Freire-Pritchett et. al). We have now included this context in the results:

Results (pgs 5-6): We next examined the types of chromatin states that were statistically significantly paired together (Fisher's Exact $p < 0.05$) and found two subnetworks, one with active chromatin states and the other with repressed or bivalent chromatin, which were discrete in iPSC-CMs (Figure 2E) and crossed over through the bivalent states in iPSCs (Figure S1H). This crossover, which is only present in iPSCs, is consistent with the role of bivalent and polycomb chromatin in pluripotency⁴⁰⁻⁴², the role of bivalency in maintaining stem cell region connectivity⁴², and with the shift of active states to bivalent and polycomb during differentiation and chromatin rewiring⁴³. This result suggests that these specialized roles of bivalent and polycomb chromatin extend to the fine-scale aspects of chromatin architecture, including loops.

Bottom of p.4. Do the authors specifically mean to say that Polycomb-mediated loops are larger and more complex than active loops or do they mean any non-active loops overall? Would be good to clarify.

We agree that these analyses were not stated clearly. We felt this wasn't a major point and due to the additional length from new analyses in response to reviewers, we have now removed these figures and the sentences.

p.5. It's confusing to mention edgeR and then proceed to describe log-CPM-based analysis. I would only mention it further down, when raw count-based analysis is introduced.

We apologize for the confusion caused by not specifying that the log-CPMs were also obtained via edgeR. We have updated the results as follows:

Results (pg6): For all loops, identified in either one or both cell types, we first compared the total normalized contact frequency (\log_2 counts per million, **logCPM**, obtained via edgeR) of the interactions between both cell types.

p.5. TMMs need disambiguation in the Results section.

We have now explained the TMM abbreviation at its first use in the results as follows:

Results (pg6): (edgeR glmQLFit on Trimmed Mean of M values, TMMs, $q < 0.01$; Figure 3B right).

p.5. What is the authors' interpretation of the fact that only iPSC-CTAL, but not CM-CTAL anchors are enriched for differential active enhancers?

We have added interpretation for these results in the context of the Freire-Pritchett paper that was referenced above, and thank the reviewer for the suggestion and reference. We believe these results are consistent with the observation by Freire-Pritchett that active elements become repressed during differentiation and chromatin rewiring; our observation that CM-CTAL anchors were not enriched for differential active enhancers would be consistent with CM enhancers being active in both cell types. This inactivation during differentiation

would result in enhancers that are present in both iPSC-CM and iPSC (i.e. non-differential), and iPSC enhancers that are not present in iPSC-CMs (i.e. differential), but not enhancers in iPSC-CMs that are not present in iPSC. We have updated the results as follows:

Results (pg7): These enrichments suggest that CTALs capture cell type specific chromatin dynamics, and are consistent with active elements shifting to repressed elements during differentiation and chromatin rewiring⁴³ (as enhancers from fetal heart tended to be present in both cell types, but enhancers in iPSCs tended to be iPSC specific). We also observed that iPSC-CTAL anchors which overlapped iPSC bivalent enhancers were more likely to overlap fetal heart bivalent enhancers (Figure 3H, blue), but not the converse, consistent with the repression of active regions of loops during differentiation, and specific use of bivalent chromatin in iPSCs^{40,41,43}. Overall, these findings show that CTALs were enriched for cell type specific functional and regulatory regions

p.6. “We first examined whether loops which colocalize iPSC-eQTLs (previously identified from a cohort including these individuals³²) to the genes that they were statistically associated with (eGenes) had stronger contact intensities within iPSCs than iPSC-CMs. We found a strong enrichment (Mann Whitney-U $p \sim 1 \times 10^{-293}$) for iPSC contact intensity above non eQTL-eGene loops (Figure 4A), indicating that loops with higher strength in a cell type may be more likely to harbor functional genetic variation”. It seems that the first sentence introduces a different kind of analysis to the one presented in the second one: the first one mentions tissue-specificity, while the second one seems to suggest comparison with non eQTL-eGene loops within a single cell type. This may just be a matter of wording.

We apologize for the confusion. We indeed meant to state that we observed an increase in the ratio between iPSC and iPSC-CM contact intensities, but instead inadvertently stated we measured only iPSC contact intensity. We have updated the results as follows:

Results (pg7): We first examined whether loops which colocalize iPSC-eQTLs (previously identified from a cohort including these individuals³⁶) to the genes that they were statistically associated with (**eGenes**) had stronger contact intensities within iPSCs than iPSC-CMs. We found a strong enrichment (Mann Whitney-U $p \sim 1 \times 10^{-293}$) for increased iPSC:iPSC-CM contact frequency ratio above non eQTL-eGene loops (Figure 4A), indicating that loops with higher contact propensity in a cell type may be more likely to harbor functional genetic variation.

p.9. Unclear sentence: “For these analyses, while data is pooled across loci to determine a genome wide trend, each pair of fold changes of phenotype and loop strength are not affected by the aforementioned biases to Hi-C read depth, as these are held constant when examining the same set of loop anchors across haplotypes”.

This sentence was originally added to remove confusion in how analyses were performed; as it seems to have only added confusion, and the full description for analyses are in the methods, we have removed this sentence.

Reviewer #2 (Remarks to the Author):

Decision:

Major revision.

Summary:

The authors seek to address a fundamental question regarding the 3D genome: do DNA mutations affect the 3D genome and to which extent? Until now, only a limited number of studies showed a link between mutations such as SNPs and 3D genome: Rao et al. (Cell 2014) using 1kb resolution Hi-C data from one cell line, Tang et al. (Cell 2015) using CTCF/RNA POL II ChIA-PET data from 3 patients and Mumbach et al. (Nature 2017) using H3K27ac/CTCF HiChIP on some cell lines. Others showed the role of structural variants on 3D genome (Lupianez et al., Cell 2016).

Here, the authors generated a resource of phased Hi-C contact maps in induced pluripotent stem cells (iPSCs) and in iPSC-derived cardiomyocytes (iPSC-CMs) from 7 patients, for which genome sequence (50X WGS), gene expression (RNA-seq) and enhancer activity (H3K27ac ChIP-seq) data were also generated (Benaglio et al., BioRxiv 2018). The authors then use allelic imbalance to assess the effect of SNPs, CNVs and imprinting on chromatin looping, as well as expression and enhancer activity. Among the contributions, the authors show quantitative differences in loop strength across cell types, instead of loop presence/absence depending on cell type. The authors also found that small changes in loop strength is associated with large changes in gene expression and enhancer activity. Finally, the study revealed that a strong impact of CNVs and imprinting on 3D genome, unlike SNPs.

Overall, the study provides novel insights into the genetic determinants of the 3D genome and its link with gene expression and enhancer activity, as well as a useful 3D genome data resource for further investigations. Moreover, the article is clear and well written.

Major revision:

The Hi-C allelic imbalance results presented here are weak due to the improper use of statistical test to rigorously assess allelic imbalance. Since Hi-C allelic imbalance results represent the core of this study, this is an important issue. To assess Hi-C allelic imbalance, the authors "calculate a Z score via a binomial approximation to a normal distribution" (1st paragraph, p29). There are two main problems with this very basic test. First, this is an approximation and thus the p-value might not be accurate when the count are low. This is problematic here because the Hi-C data are typically sparse and the counts are often low. Note that the binomial exact test should be used instead of the normal approximation test when no overdispersion is found in the data. Second, NGS counts are known to be overdispersed and thus do not follow a binomial law. Not accounting for overdispersion might represent a big issue, since it might lead to many

false positive variants. To account for overdispersion, different tests were proposed that overcome the binomial exact test and account for overdispersion, that are implemented for instance in WASP [1] and MBASED [2] or any other dedicated program. The authors should use a program dedicated to assess allelic imbalance that will account for overdispersion and biases. It is likely that the authors will find a smaller number of significant Hi-C allelic imbalances (HTALs) using a dedicated program than detected using the normal approximation (114 HTALs). If the authors detect only a few number of significant HTALs using a dedicated program which therefore do not allow them to integrate with other data, and thus prevent further key analyses in the article, I believe the authors should keep the normal approximation test they used but they should clearly state in the article the weakness of the test they used (and thus many false positives might be detected), explain why they used it instead of a test accounting for overdispersion and that the results should be interpreted with caution. Another option might be to take an FDR=10% or 20% and keep a lower number of loops for testing (only the strongest loops above a certain threshold) to reduce p-value correction effect.

We agree that the binomial test could be affected by overdispersion in the read counts and have now explored the extent that it affected our results. We utilized the WASP package to estimate overdispersion in Hi-C reads and calculate ASE per individual with a beta-binomial test. After meta-analyzing these p-values with Fisher's Method, we identified 89 HTALs, rather than 114. These 89 HTALs were enriched for imprinted loci at an even higher rate than the 114 loops identified from the half-normal distribution and showed overall similar results for the analyses that we performed in Figure 6 for HTALs. As the results and interpretations were broadly equivalent between the two methods, we chose to keep the half-normal approach in the main text, but have now added descriptions of these new analyses in the Results, Methods, and as Supplemental Figure 7.

Results (pg10): Next, for each individual, we identified imbalance via a Z score using a half normal distribution (see Methods for details and for complementary analysis using WASP), following which we combined the p-values across individuals with Fisher's method for meta-analysis.

Methods (pg31): To identify HTALs with a beta-binomial test, we utilized the combined haplotype scripts from WASP. First, we created a CHT input file using the haplotype counts for each loop. Next, we passed these files to `fit_as_coefficients.py` to calculate the binomial overdispersion parameters. Finally, we obtained p-values for each individual separately from `combined_test.py` with the option `-as_only`. These p-values were combined via Fisher's method and both the combined and raw p-values were used for downstream analyses. We repeated the analyses from Figure 6 using these results and observed similar enrichment patterns to the half-normal approach (Figure S7).

Reviewer #3 (Remarks to the Author):

This paper describes interesting findings about the role of changes in quantitative loop strength in biological outcome. This is important work, as it challenges the assumption that only complete loss or gain of loops will be biologically meaningful, and it

complements previous observations that changes in major architectural factors have sometimes surprisingly subtle effects on genome organization. However, given that the study places extensive emphasis on subtle and quantitative changes in loops, I am concerned about some of the steps taken to define loops. Admittedly, loop calling is a hard problem with no one right answer. But I am concerning that they tuned their loop-calling algorithms to focus on the strongest “dots” in the Hi-C heatmap, presuming that these are the most biologically functional and gene regulatory loops, when these may be the most invariant, primarily architectural features. Also concerning is the pooling of individuals or cell types, which may blur out specific loops that would have been observed in specific samples. There are also places in the haplotype section where the writing of the manuscript becomes unclear and hard to follow, and could use revision.

My detailed comments are below:

The authors state “As chromatin structure has been shown to be an evolutionarily stable trait, changes in chromatin loops across cell types are a priori more likely to take the form of subtle strength modulations than the formation of entirely new loops” This is an interesting point, and it is indeed worth noting that changes may be subtle, but this ignores the different levels of variation at different scales of chromatin structure: A/B compartments change notably across cell types, and small scale chromatin structure (local compaction and accessibility) likewise varies, as may local enhancer promoter connections. It is mostly the CTCF-mediated TAD-scale loops that are stable across syntenic regions and cell types. The authors should be careful about their definitions and language: if they mean certain types of loops, then they should not imply this encompasses all chromosome structure.

Through our efforts to frame chromatin loop strength in a more biologically meaningful lens as asked by reviewer 1 and by reviewer 3 at a later point, we have reframed/heavily edited the first paragraph of the introduction. Due to this process, we no longer discuss the evolutionary stability of chromatin architecture.

Additionally, we have also made changes in in response to reviewer #1's comment 1 to be more explicit about the types of loops that we analyze (please refer to these comments for more detail). Using existing promoter capture datasets from iPSCs and iPSC-CMs and CTCF ChIA-PET data from GM12878, we show that the loops that we analyze show strong overlap with promoter-based loops and structural loops. We also extended our functional analyses where we examined the enrichment of promoter enhancer loops to HTALs (Figure 6E) to all pairs of chromatin states, as well as pHic and CTCF ChIA-PET, and show that the loops none of these overlaps were statistically significant (FDR $q > 0.05$). We have included these analyses in the results, and hope that these additional changes provide a more explicit characterization of the identified loops and HTALs.

Results (pg 10): We compared the proportion of the 114 HTALs that were also iPSC-CTALs, CM-CTALs, iPSC called, or iPSC-CM called loops to the corresponding proportion of union loops.

However, we found no significant differences for any association ($p > 0.05$ for all tests; Figure 6A). We next examined whether a particular type of loop was enriched within HTALs (ie CTCF loops, promoter-enhancer loops; **see Methods**), and found no significant enrichment (FDR $q > 0.05$); together, these results indicate that loops which varied between haplotypes were not more likely to be a specific type of loop.

Methods (pg 32): **Chromatin state enrichments at HTALs**

To examine whether any pairs of chromatin states were enriched at opposing HTAL anchors, we annotated all HTAL anchors with the chromatin states they overlapped (with iPSC or fetal-heart chromatin states) via pgltools intersect1D. Next, we used a Fisher's Exact test for each pair of states (125 pairs total) to compare the proportion of HTALs with the states at their anchors to the proportion of non-HTALs. Finally, we FDR corrected the p-values for multiple-testing control.

Loop set enrichments at HTALs

To examine whether any loop sets (CTALs, cell type called loops, CTCF ChIA-PET interactions, or pHiC interactions) were enriched for HTALs, we annotated HTALs by the loops they overlapped via pgltools intersect. Next, we used a Fisher's Exact test for each loop set.

I am concerned about the justification for pooling Hi-C data from different individuals for the same cell type. The Pearson correlations are calculated between samples with 1 Mb bins. First, it has been shown that Pearson correlation is not a good way to evaluate Hi-C similarity (Yaffe and Tanay, 2011; HiCRep tool, Yang et al, 2017; Yardimci, BioRxiv, 2017). Second, even if Hi-C datasets are similar at 1 Mb resolution, there is no guarantee that they are similar at loop-scale (10-500 kb resolution). I understand that high numbers of reads are needed for loop calling, so pooling more reads could increase resolution and power, but if the whole argument is to look for subtle changes in loop strength, isn't it quite likely that some subtle changes will be averaged out across individuals? I suggest that the authors more deeply sequence 1 individual of each cell type and check how the called loops of that single individual compares between cell types and to the pool of the other individuals.

We agree that using a tool that was built for testing reproducibility across Hi-C data would be better than the general Pearson correlation. We have now implemented HiCRep in addition to Pearson correlation and find similar results. We have included this analysis in Figure S1B, and have updated the Results as follows:

Results (pg4): We performed hierarchical clustering of the contact frequencies by cell type across individuals and observed high correlations within each cell type both by Pearson correlation and correcting for Hi-C biases via HiCRep³⁷(Figure S1B).

We additionally apologize for confusion that was caused from our wording surrounding the cell type data pooling as it was not clear that we used the pooled data only to identify loop locations. Unlike analyses for differential gene expression in which genic regions are well defined, differential loop strength/peak heights first require defining a set of loops/peaks where signal will be examined consistently across individuals. We used pooled data across individuals for loop

calling in order to gain power and identify a larger number of “reference” loops in order to obtain the most comprehensive set of regions for quantitative analyses of signal on a per individual/cell type basis. In order to identify differential signal at these loops, we then examined Hi-C and molecular phenotype signals within each individual and examined general trends across cell types or haplotypes. We also used the individual sample data with edgeR in order to identify CTALs, and quantified ASE was per individual in order to identify HTALs before performing a Fisher’s method meta-analysis. We have clarified these points in the results as follows:

Results (pg5): To identify a set of reference loops for downstream quantitative analyses, we combined the Hi-C data within each cell type to obtain a comprehensive set of loops from high-depth data. We pooled the data across samples for each cell type, resulting in reference chromatin maps with the highest resolution (~2kb matrix resolution, defined as the resolution at which 80% of loci have 1000 or more contacts with any other locus²) in iPSCs and iPSC-CMs (or any other iPSC derived cell type) to date and were comparable in resolution to the Hi-C map in GM12878² (Figures 1C & S1C).

Finally, we examined whether subject-specific cell type associations were averaged out in the CTAL analysis. To examine this, we stratified the relationship between iPSC and iPSC-CM CPMs across cell types by individual, created a figure for each, and have now included these as Figure S2. Within individual subjects, we recapitulated similar patterns as those in the averaged data: loops were generally consistent across cell types and did not have a high contact intensity in one cell type, and low in the other. Notably, there is a banding pattern at low contact frequencies where loops appear to have higher contact intensity within one cell type, but these only occurred at loops with a low contact frequency and maintain the same distribution/shape at the higher contact intensity loops, suggesting that this resulted from noise due to the lower read depth at these loops overall. These findings suggest that the combining of the data did not prevent the analysis of loops that only occurred in one cell type in one individual. We have updated the results to reflect these changes as follows:

Results (pg6): For all loops, identified in either one or both cell types, we first compared the total normalized contact frequency (\log_2 counts per million, **logCPM**, obtained via edgeR) of the interactions between both cell types. We observed that the majority of loops that were called in both cell types (grey in Figure 3B left) had high logCPMs in both cell types, whereas the loops that were only called in a single cell type (blue or red in Figure 3B left) tended to have overall low logCPMs and often showed highly similar contact intensities between cell types. We did not observe, however, loops with a high logCPM in one cell type, and a very low logCPM in the other. These patterns were similar within subjects, suggesting that these subtle modulations in logCPM across cell types were not due to the combination of data across individuals (Figure S2). These results indicate that chromatin loops that were called as differentially present or absent between cell types were often of low logCPM, and were therefore likely to be inconsistently identified by the loop calling algorithms. Thus, the differences in the loop sets between the two cell types were not due to the establishment of novel loops present in only one cell type.

Results (pg9): Within each cell type, for each individual, we assigned informative Hi-C contacts carrying a phased allele to each haplotype (Figure 5A) and examined allelic imbalance across all

loops. Next, for each individual, we identified imbalance via a binomial test using a half normal distribution (see methods), following which we combined the p-values across individuals with Fisher's method for meta-analysis.

Calling loops at multiple resolutions and with multiple algorithms is a great idea, but I am not sure it is justifiable to use "visual best match" as the criteria for whether loop calling is working well. It is an oversimplification to state that "Fit-Hi-C is known to detect false positive loops along TAD boundaries" Why should these be called false positives? These lines along boundaries are likely due to loop extrusion and represent real enrichment of interactions between one boundary (loop anchor) and other regions that slide past. These may be biologically meaningful interactions. On the flip side, we don't know that the prominently visible "dots" in Hi-C maps (assumed here to be "true positives") are actually more meaningful. Indeed, these TAD corner peaks may be the most invariant and thus least likely to relate to changes in gene expression. So, I'm concerned that the authors may have artificially tuned their algorithms to detect visually obvious peaks while unjustifiably excluding others. If this kind of "visually correct" filtering is not done, would there be more correlation with gene expression differences and called peak changes?

We apologize for the overstatement about false positive loop calls from Fit-Hi-C, and for the confusion around why we filtered loop calls. While our method did decrease the number of loop calls that were on TAD boundaries, it also reduced the number of loops that were far off the diagonal and clearly background signal. Additionally, we have performed some analyses to examine how filtering these loops affected overlap with previous loop sets. For both the loops called by HICCUPS and by Fit-Hi-C, we examined the percent of loops that were present in GM12878 or in Promoter Capture Hi-C as a function of number of resolutions, and found that as we filtered loops, a higher proportion of our loops overlap these previous studies, and include these analyses in Figure S5D. We have addressed this by making the following changes in the methods:

Methods (pg28): Loop calling techniques are known to be technically variable³⁸. We found many loop calls from both Fit-Hi-C and HICCUPS that were located at random points throughout the Hi-C matrix far off the diagonal (Figure S6C). We thus developed a procedure to remove these loop calls by examining the number of resolutions at which the loop was identified. We intersected loop calls across all resolutions within each calling method, retaining the highest-resolution call at each intersection event, and filtered out loops present in less than 3 or 7 resolutions for HICCUPS or Fit-Hi-C, respectively. The loops retained in these filtered sets visually appeared to best represent the underlying Hi-C data (Figure S6C). Next, we compared how these filtered sets overlapped with promoter-capture HiC⁷² or the Rao. et al loop set and found that using these filtering criteria resulted in a higher overlap with the retained loops (Figure S6D), suggesting that this filtering strategy removed spurious loop calls. After this filtering, while we found a large number of loops that overlapped between Fit-Hi-C and HICCUPS (Figure S6E), many loops were unique to only one caller (Figure S6F). We therefore intersected the loops across calling methods, retaining the loop with the smallest total anchor size at each intersection event (Figure S6G). Overall, this process retained the smallest resolution loop call for all loops

present in either 3 HiCCUPS or 7 Fit-Hi-C resolutions, and resulted in iPSC called and iPSC-CM called loop sets.

To address whether our loops were filtered to only those at TAD boundaries, we called TADs using the Dixon et al. method and overlapped TAD boundaries with our loop calls. We found that only 2.9% and 5.1% of iPSC and iPSC-CM loops had both anchors within 25kb of TAD boundaries. We thus do not believe that focusing on these sets of loops is looking at only structural loops at TAD boundaries. We have added these analyses to the Results and Methods:

Results (pg5): To examine whether these loops were predominantly demarcating TADs, or were separate from TAD structure, we called TADs in both cell types and examined the number of loops that had both anchors within 25kb of TAD boundaries. We found only 2.9% of iPSC loops, and 5.1% of iPSC-CM loops, to have both anchors at TAD boundaries, indicating that these loops were not only those demarcating TADs.

Methods (pg28): **Identification of TADs**

To identify TADs, we utilized the HMM method from Dixon et. al 2012 {Dixon, 2012 #53} with the HiC matrix at 40kb resolution as recommended. To determine the percent of loops that were at TAD boundaries, we paired TAD boundaries sequentially in the file to create a pgl format file, and then used pgltools intersect to find the percent of loops with both anchors at TAD boundaries.

The idea of using raw reads from all replicates at each loop to define differences between cell types is nice, and it is a key observation that loops not “called” in one cell type often still have similarly high contacts around that region. But I’m concerned that by looking at raw reads just at the loop location (and nothing surrounding it), a presumed difference in “loop strength” could in fact be a difference in overall interaction decay with distance between the samples (due to an overall A/B compartment or compaction change, for example). Relatedly, Figure 2A shows a very nice example of cell-type specific differences, but reducing this to the few called loops seems incomplete: iPSCs have a whole domain structure that is not visible in CM cells.

We understand the reviewer’s concern that pinpoint changes in chromatin architecture (ie at loops) could be due to changes in a larger scale. The local relationship between the increased contact intensity at loops and increased expression or regulation is biologically interesting, but we agree these larger changes should be examined to ensure they are not driving the local effects. We have examined how changes external to the looped loci are associated with CTAL status by examining the relationship between differential contact strength and A/B compartments. After identifying compartments via juicertools, we examined whether A compartments have stronger intensities than B compartments within the same cell type, and found a statistically increased contact intensity within A compartments. We thus next examined the relationship between contact intensity across cell types and differential compartments across cell types by finding the percent of variance explained between CPM fold change

and PC difference. We found an r^2 of 0.009, which indicates that while CPM differences and PC differences are related, differences in compartments explain a small fraction of differential loop strength, and thus are unlikely to drive CTALs. To confirm this finding, we subset the loops by the types of compartments both anchors fell in in both cell types, and found that the CTAL trend was consistent across all types of compartment cell-type combinations we observed. We have now integrated this analysis into Supplemental Figure S2 and the Results:

Results (pg 7): To determine whether 3D architecture at a compartment level contributed to these differences, we identified A and B compartments² and partitioned the loops by their location in both cell types. While we found increased contact propensity within A compartments relative to B compartments in both cell types (Figure S3A), the percent of variance in logCPM explained by compartment differences was only 0.009 (Figure S3B). Additionally, we found that the CTAL distribution was consistent across all types of anchor-compartment-cell type combinations (Figure S3C). These results suggest that compartment differences did not drive CTALs.

Methods (pgs 27-28):

Identification of Compartments

Chromatin compartments were called for each cell type via Juicer command line tools using the corresponding .hic files where the first PC of the normalized contact frequency matrices were extracted at 1Mb resolution. The signs of the PC eigenvectors were used to stratify each chromosome into two arbitrary compartments. To determine the activity status of the two compartments on each chromosome, we counted the number of reads from 1) RNA-seq, 2) H3K27ac ChIP-seq, and 3) ATAC-seq aligned to each of the 1Mb bins from all available samples for each cell type, average the read counts across all samples for each assay in each cell type, and assigned the compartment with higher average read counts from all the three assays as the active compartment (A) and the other compartment as inactive compartment (B). While most of the time all three assays had consistent compartment activity calls, chr21 of iPSC and chr22 of iPSC-CM had inconsistent calls, where we assigned the compartment activity based on the majority of assays.

What about checking the profile of iPSC H3K27Ac or CTCF at CM-specific peaks? Is there indeed less enrichment?

We agree that examining the enrichment of H3K27AC and ATAC-seq at CTAL anchors in a cell type dependent manner would help validate our quantitative identification of cell-type relevant loops and thank the reviewer for the suggestion. We have now examined the tag-distributions of both data types on CM-CTAL, non-CTAL, and iPSC-CTAL anchors, and found these three anchor types to be enriched cell type specifically as expected (ie iPSC H3K27AC was enriched at iPSC-CTAL non-CTAL anchors above CM-CTAL anchors). We have added these analyses to the Results and included a new figure panel (Figure 3G):

Results (pg7): We examined how the enrichment of H3K27ac and ATAC-seq signals varied by CTAL status, and found that within each cell type, CTALs of that cell type, and non-CTALs, had the highest H3K27AC and ATAC-seq signal, while CTALs of the other cell type were least enriched (Figure 3G). These enrichments suggest that loops with decreased contact propensity may be less likely to be involved in gene regulation despite being present in the cell.

Page 5: There are references to figure 2G, but this figure doesn't exist. Should this be 3G? Either way, I can't see how the statements made are supported by the figure (where are the regions enriched for heart and iPSC bivalent chromatin? That can't just be the generic category 12_EnhBiv...

Thank you for catching the typo on Figure 2G; we did in fact mean Figure 3G. Additionally, we have updated the text surrounding to be more specific about bivalent chromatin as follows:

Results (pg7): We also observed that iPSC-CTAL anchors which overlapped iPSC bivalent enhancers to be more likely to overlap fetal heart bivalent enhancers (Figure 3H, blue), but not the converse, consistent with the repression of active regions of loops during differentiation, and specific use of bivalent chromatin in iPSCs^{40,41,43}.

It would be well for the authors to consider, and adjust their language accordingly, what is actually meant by “loop strength”. This is likely a measure of % of the cell population that had the loop, which probably should at least once be clarified to mean “loop frequency”. The number of contacts does not tend to directly correspond to “the distance in 3D space between two anchors” as claimed on page 7.

We agree that a more biologically motivated explanation for loop strength would strengthen our manuscript. Additionally, we have thoroughly reviewed literature discussing the relationship between contact intensity and 3D distance, and found that while the two are not equals measures, FISH experiments have shown a monotonic relationship between 3D distance and contact frequency at loops, and polymer physics models have been employed to reconstruct the physical 3D shape of the genome from Hi-C data. We have included these concepts, and re written our intro and discussion as follows:

Introduction (pg3): Chromatin loops colocalize regulatory elements with their targets¹⁻¹⁵ by bringing genomic regions that are distant from one another in primary structure close together in 3D space¹⁶. These colocalized regions, also known as loop anchors, are preferentially enriched for disease associated distal regulatory variation and expression quantitative trait loci (eQTLs)¹⁷⁻²². While it has been shown that the physical 3D distance between looped loci can vary^{16,23-25}, previous studies examining cell type and haplotype differences in looping have considered loops to be either present or absent, rather than as a quantitative phenotype. Thus, the extent to which quantitative differences between chromatin loops exist, and whether they are associated with differences in gene expression and regulation, has yet to be explored. Moreover, the extent that regulatory genetic variation acts by affecting the physical distance between looped loci is not yet known.

Bulk chromatin conformation assays (e.g. 3C, 4C, and Hi-C) were designed to identify genomic regions in close spatial proximity¹⁶ by measuring physical contact frequency between two pieces of colocalized (ie looped) DNA in a pool of cells. As an individual contact is a physical interaction in a single cell, the contact frequency measured in a pool of cells reflects the probability for the interaction to occur (**contact propensity**) across all cells in the sample. Overall contact frequency is a function of spatial proximity, which has enabled Hi-C data generated in either bulk²⁶⁻²⁹ or single cell³⁰ assays to be used to reconstruct the physical 3D structure of chromatin at the compartment, TAD, and loop levels by incorporating polymer physics models. Additionally, single cell Hi-C has shown that while single cell contacts occur at loops called from bulk data, individual cells vary in the predicted physical distances between their looped loci³⁰. Moreover, fluorescence in situ hybridization studies have shown that spatial distance and contact frequency between looped loci vary concordantly²³⁻²⁵. Together, these studies imply that differences in contact propensity reflect differences in the physical distance between the looped loci. Therefore, investigating contact frequency as measured by Hi-C, in combination with molecular phenotypes, may reveal if the physical distance between looped loci varies across cell types, and if this is associated with differential regulation of gene expression.

Discussion (pg12): We performed quantitative comparisons of contact frequency across cell types and haplotypes to identify differences in chromatin looping, and integrated these differences with quantitative measures of differential expression and H3K27ac to examine the functionality of contact propensity. These analyses revealed a proportional association between contact frequency and gene expression/H3K27ac, which surprisingly linked the phenotypes across different magnitudes of variability: extremely subtle changes in contact frequency were associated with large differences in gene expression and H3K27ac. If contact propensity at loops is a fundamental regulator of gene expression, differences in contact propensity would be expected to be associated with similarly sized differences in gene expression regardless of the environment in which the differences occurred (ie across cell type, haplotype, or experimental conditions). As we observed a consistent relationship between the two, we believe these data indicate that contact propensity is a mechanism involved in regulating gene expression, similar to enhancer activity or transcription factor binding strength. 3D modeling of the genome from Hi-C^{26-30,52}, and fluorescence in situ hybridization experiments^{23,25,53}, have shown a relationship between contact frequency and spatial proximity, which suggests that loci with increased spatial proximity come into physical contact more easily, and thus produce a Hi-C contact more readily. Extending our results to spatial proximity, our data suggests that genetic variation at chromatin loops may exert their effects on gene expression or gene regulation by modulating the spatial proximity between the loci of, and thus contact propensity at, chromatin loops. Notably, as we identified a non-directional correlation, contact propensity may either affect, or be affected by, gene expression and/or regulation. In the latter case, genetic variation would mediate effects on spatial proximity through affecting gene expression or regulation.

It is worth noting that previous work has suggested that the degree of difference between haplotypes may vary across cell types (Rivera-Mulia, 2018 shows a good example of this in replication timing: ES cells have more variation between haplotypes than differentiated cell types) So, this makes it a little dangerous to pool Hi-C data between iPSC and CM for haplotype comparison. Also, it then seems a circular argument to pool cell types to call HTAL, but then claim to find that haplotype variable loops are not cell type variable.

(Didn't the pooling of cell types make it much harder to identify haplotype variable loops that also vary between cell types?)

We agree that ASE can be different across cell types (or, in the example provided by the reviewer, within different cells in the same cell type due to a process such as replication timing). In order to examine whether it was safe to combine read estimates to gain additional power for HTAL identification, we examined the correlation in the maternal allele frequencies between iPSCs and iPSC-CMs at HTALs identified using either only the iPSC data, or the iPSC-CM data (Supplemental Figure 5). We found that the maternal allele frequencies were highly correlated, and thus combined the data. We have made this clearer in the text by making the following changes:

Results (pgs9-10): As we observed that the maternal allele frequencies were highly correlated across cell types, to increase power for these analyses, for each of the 26,679 chromatin loops in the union set, we pooled contacts for each individual across their corresponding iPSCs and iPSC-CMs.

We agree that it appears circular to pool the cell types and then state that HTALs are not enriched for a particular cell type, despite finding highly concordant imbalance across cell types. We have now examined whether the cell type called HTALs are enriched within CTALs for either cell type and found that neither set were enriched. We have included this in the manuscript as follows:

Results(pg9): This process identified 54 total HTALs: 27 from iPSCs, and 27 from iPSC-CMs. We first examined whether these 54 HTALs were enriched for being CTALs of either cell type and found no significant enrichments (Fisher Exact $p > 0.05$).

The last two sentences on page 8 are very confusing, possibly due to errors: "this observation is consistent with linearly closer loci having increased Hi-C contact strength, and thus more power for detection, as deletions reduce the linear space between loci increasing contact strength thereby increasing power, and duplications increasing the space increasing contact strength thereby decreasing power. Overall, these results confirm that genetic imprinting, and suggest that CNVs, may be strong drivers of allelic imbalanced chromatin looping." How can both reducing and increasing the space increase contact strength? What do you mean "results confirm that genetic imprinting"? Is it fair to infer that raw contacts indicate loop strength among regions whose linear distance has been changed? Don't you need to compare the loop contacts to surrounding contacts to see if the relative loop strength has actually changed?

We apologize for the oversight in the wording in this sentence; both deletions and duplications would not increase contact intensity. Duplications would decrease frequency, and deletions would increase frequency. We fixed the CNV sentence and clarified our meaning of "confirm that genetic imprinting" as follows:

Results (pg10): Thus, while CNV type is not associated with allelic imbalance, loop detection may be affected by CNV presence. The observed pattern of enrichment in deletions is consistent

with linearly closer loci having increased Hi-C contact strength as deletions reduce the linear space between loci, thereby increasing contact frequency; conversely, duplications increase linear distance and thus decrease contact frequency. Overall, these results confirm previous reports which suggested that genetic imprinting^{2,5} may be a strong driver allelic imbalance, and suggest that CNVs may have smaller effects on allelic imbalance in chromatin looping

Additionally, we agree that CNVs could appear to be enriched in loop calls or for allelic effects without actually causing these effects due to their enlarging/shrinking of the linear genome. To fully test if the effects seen are due to this artifact, methods of personalized CNV phasing need to be developed. We have clarified this possibility in the Results as follows:

Results (pg10): Thus, it is unclear how much of this enrichment is due to a technical artifact induced by increased contact frequency from a linear genomic distance change.

Minor points:

Fig S1A refers to “regular read pairs” and “normal split read pairs”. It is not clear what “regular” means here, nor is that defined on the Juicer output or documentation, so this should be clarified.

We apologize for the confusion created from these terms. Normal read pairs are paired end reads in which both pairs fully map to one restriction enzyme cut site. Normal split read pairs are where one or more reads are chimeric (ie map to multiple cut sites), but the insert size of the fragment is still within the expected insert size. We have updated the supplemental figure legends to clarify this as follows:

Supplemental Legend Figure S1: Normal read pairs are those where both ends map to a single restriction enzyme cut site. Normal split read pairs are those where one or more end maps to multiple restriction enzyme cut sites, but the insert size of the read is as expected. Abnormal split read pairs are those where one or more end map to multiple restriction enzyme cut sites, but the insert size of the ends is larger than expected.

Fig S1C is a nice graph, but I’m very surprised that nearly 100% of all 5 kb bins would have 2000 contacts per bin, even in the interchromosomal interaction space. Does this set of bins truly include all contact bins in the genome? Or is it the sum of contacts involving every given bin?

We apologize for the confusion. The reviewer is correct that the resolution measurement is the total number of contacts that have one end within a bin (ie the sum of contacts involving every bin and a particular bin). This is the definition given by Rao. et. al. We have updated the main text to clarify what resolution means as follows:

Results (pg5): We pooled the data across samples for each cell type, resulting in reference chromatin maps with the highest resolution (~2kb matrix resolution, defined as the resolution at

which 80% of loci have 1000 or more contacts with any other locus²) in iPSCs and iPSC-CMs (or any other iPSC derived cell type) to date and were comparable in resolution to the Hi-C map in GM12878² (Figures 1C & S1C).

Fig S1 legend has a typo: “edges connecting statistically significant pairs of chromatin states found at opposing.” (opposing what?)

We have updated the text to:

Supplemental Legend: edges connecting statistically significant pairs of chromatin states found at opposing anchors.

Make sure Fig S4C is a high resolution enough figure for readers to see the pink and blue squares you're drawing (I can't make them out)

We apologize for the low resolution figure; we have increased the resolution so that one can zoom in on the PDF and clearly see the squares that are drawn in, and have updated the legend to reflect this:

Supplemental legend: Hi-C heatmaps with loops shown for each caller at different filtering criteria. These images are high resolution; zooming on the PDF is encouraged.

Page 4: MEME should be capitalized

We have capitalized MEME throughout the manuscript.

Page 7: “ASE genes/peaks” are not defined (what does “ASE” stand for?) and make subsequent paragraphs hard to read

We have added a disambiguation for ASE at its first occurrence as follows:

Results(pg6): Next, we assigned informative reads from H3K27ac and RNA expression to each individual's maternal or paternal haplotype using MBASED⁴⁰, and then identified significant peaks or genes with allele specific effects (**ASE**; FDR $q < 0.05$) within each individual using a binomial test.

Reviewer #1 (Remarks to the Author):

The authors have clearly put considerable thought and effort in revising the manuscript, and it is now much improved. My main remaining concern, shared by reviewer 3, is whether minor changes in loop strength are interpretable/interesting quantities. However, the authors have caveated their findings appropriately in the revised version, and perhaps it's worth getting these data out and letting the community decide.

I would just ask that WASP results be referenced more prominently in the Results, with the differences in the numbers of significant HTALs stated explicitly. I agree however that the benefit of swapping to WASP results throughout the study would unlikely justify the time and effort required for this at this late stage.

Reviewer #2 (Remarks to the Author):

The authors have substantially improved their manuscript regarding allelic imbalance testing.

The authors wrote that "We utilized the WASP package to estimate overdispersion in Hi-C reads and calculate ASE per individual with a beta-binomial test. After meta-analyzing these p-values with Fisher's Method, we identified 89 HTALs, rather than 114. These 89 HTALs were enriched for imprinted loci at an even higher rate than the 114 loops identified from the half-normal distribution and showed overall similar results for the analyses that we performed in Figure 6 for HTALs. As the results and interpretations were broadly equivalent between the two methods, we chose to keep the half-normal approach in the main text, but have now added descriptions of these new analyses in the Results".

If the results are similar between the half-normal distribution test and betabinomial test, the authors should show that the SNPs found by both tests strongly overlap.

In my opinion, if this is not the case, the authors should include the results from the beta-binomial test in the main text, and put the results from the half-normal test in the supplementary.

Reviewer #3 (Remarks to the Author):

The authors have done well to address nearly all the points that I and the other reviewers raised during this revision. In particular, I appreciate the way that they clarified the loop calling and pooling approach that they used, adjusted their language to more accurately represent contact propensity, and tested loop frequency differences between the A and B compartment.

I have a few remaining issues, which are minor, but I felt worth noting.

First, just some very small things:

1) ChIA-PET data does not come from Rao et al, as stated on page 4 of the response to reviewers. The reference does appear to be correctly cited in the text as Tang et al., 2015, but make sure Rao is not cited as the source of this data.

2) Line 879: "HICCPUS" should be "HICCUPS"

A bit more substantial issues:

1) I still don't understand the updated explanation of "Normal", "Normal split" and "Abnormal" read pairs in the legend of S1A, and I note that their definition doesn't match the figure (which uses the word "regular", which is not defined). Defining normal as "both ends map to a single restriction enzyme cut site" makes it sound like the ends map to adjacent restriction fragments, both near a single cut site. But I think maybe this means "both ends map uniquely". Then, for "normal split", do you mean that one or both ends are "multiple mapped"- having several potential locations in the genome? And then you're picking the location that gives an expected insert size? Am I correct that you're addressing the issue of unique vs. multi-mapped alignment with these terms? Such as described here: <https://pairtools.readthedocs.io/en/latest/parsing.html> ? Perhaps you're just using your own words for what the Juicer report calls Normal Paired, Chimeric Paired, and Chimeric Ambiguous? If so, why not use those words? The word "chimeric" helps me understand a bit better, but you don't use this word in your legend.

2) The authors have done well to switch to the term "contact propensity" rather than "loop strength" and to clarify what they mean in terms of looping frequency across the population of cells. But, I still think they are equating distance and contacts in a way that is not accurate, or necessary to their conclusions. Indeed, as the authors note, FISH measured distances and Hi-C contact frequency are often inversely proportional, but the strongest relationship is between contact frequency and FISH colocalization (distance less than a threshold). Oddly, as one of the references for the sentence:

"Moreover, fluorescence in situ hybridization studies have shown that spatial distance and contact frequency between looped loci vary concordantly"

the authors cite a paper by Imakaev and Fudenberg. This paper specifically finds "a nontrivial relationship between contact frequency and spatial distance" in which equivalence between the two can be easily broken by dynamic looping contacts—the very situation the authors explore here. Two loops might have the same interaction probability in Hi-C: both occurring in 10% of cells. But if, for

one loop, the two partners stay relatively close together when not touching and for the other loop, the two partners are far apart when not touching, the average spatial distance of these two loops would be vastly different.

I hate to belabor this point, because I think the author's conclusions are valid and interesting solely focusing on what they can measure: contact propensity. They do not have to draw an equivalence to spatial distance in order for their results to be biologically interesting. So, I would recommend that they just stop at the point of explaining what contact propensity is and take out some of their sentences about physical distance between loci.

3) Related to the above, I am uncertain why the authors need to invoke spatial proximity, rather than simply contact frequency, in their discussion of how genetic variation may exert its effects: "Extending our results to spatial proximity, our data suggests that genetic variation at chromatin loops may exert their effects on gene expression or gene regulation by modulating the spatial proximity between the loci of, and thus contact propensity at, chromatin loops."

I don't think the conclusion has to be extended to overall average spatial proximity to be interesting. I can see a mechanistic explanation for how variation at a specific loop site could affect contact frequency itself: a loss or gain of protein binding due to the variation may make looping interactions more or less favorable/stable. Why do you need to propose that the variant instead works by causing two loci to stay closeby even when not interacting? How would this work? That seems like a property controlled by the chromatin state of the broader region, tethering to nuclear structures, and cohesin-mediated loop extrusion.

We would like to thank the reviewers for their additional helpful comments in improving the quality of our manuscript. We have responded to each point raised by the reviewers underneath their original comments.

Reviewer #1 (Remarks to the Author):

The authors have clearly put considerable thought and effort in revising the manuscript, and it is now much improved. My main remaining concern, shared by reviewer 3, is whether minor changes in loop strength are interpretable/interesting quantities. However, the authors have caveated their findings appropriately in the revised version, and perhaps it's worth getting these data out and letting the community decide.

I would just ask that WASP results be referenced more prominently in the Results, with the differences in the numbers of significant HTALs stated explicitly. I agree however that the benefit of swapping to WASP results throughout the study would unlikely justify the time and effort required for this at this late stage.

We thank the reviewer for the suggestion and have now incorporated the WASP results more prominently in the Results section. We have also added the observation about imprinting that was present in the response to reviewers to the Methods section:

Results (pg9): Next, for each individual, we identified imbalance via a Z score using a half normal distribution (as well as using the computational framework WASP; see Methods and Figure S7 for details of complementary analysis), following which we combined the p-values across individuals with Fisher's method for meta-analysis.

Results (pg9): In comparison, we observed slightly fewer HTALs ($N = 89$) with the WASP analysis; however, the majority (83/89, 93%) were found in both sets.

Methods (pg32): To identify HTALs with a beta-binomial test, we utilized the combined haplotype scripts from WASP. First, we created a CHT input file using the haplotype counts for each loop. Next, we passed these files to `fit_as_coefficients.py` to calculate the binomial overdispersion parameters. Finally, we obtained p-values for each individual separately from `combined_test.py` with the option `-as_only`. These p-values were combined via Fisher's method and both the combined and raw p-values were used for downstream analyses. This analysis resulted in the identification of 89 HTALs, 83 of which were contained in the half normal HTAL set (93%). We repeated the analyses in Figure 6 using the WASP results and observed similar enrichment patterns to the half-normal approach, but found stronger enrichments at imprinted loci (Figure S7).

Reviewer #2 (Remarks to the Author):

The authors have substantially improved their manuscript regarding allelic imbalance testing. The authors wrote that "We utilized the WASP package to estimate overdispersion in Hi-C reads and calculate ASE per individual with a beta-binomial test. After meta-analyzing these p-values with Fisher's Method, we identified 89 HTALs, rather than 114. These 89 HTALs were enriched

for imprinted loci at an even higher rate than the 114 loops identified from the half-normal distribution and showed overall similar results for the analyses that we performed in Figure 6 for HTALs. As the results and interpretations were broadly equivalent between the two methods, we chose to keep the half-normal approach in the main text, but have now added descriptions of these new analyses in the Results".

If the results are similar between the half-normal distribution test and betabinomial test, the authors should show that the SNPs found by both tests strongly overlap. In my opinion, if this is not the case, the authors should include the results from the beta-binomial test in the main text, and put the results from the half-normal test in the supplementary.

We thank the reviewer for pointing out our oversight in not comparing the two sets of HTALs identified by the half normal and the beta binomial (WASP) approaches. We have now included this comparison, and report high overlap between the methods with 93% of the WASP HTALs being captured by the half normal approach. As suggested by the reviewer, since this overlap was quite large, we left the results as is with the half normal approach. However, we have added this comparison, as well as the interpretation of the WASP results in the Results and Methods sections as follows:

Results (pg9): Next, for each individual, we identified imbalance via a Z score using a half normal distribution (as well as using the computational framework WASP; see Methods and Figure S7 for details of complementary analysis), following which we combined the p-values across individuals with Fisher's method for meta-analysis.

Results (pg9): In comparison, we observed slightly fewer HTALs (N = 89) with the WASP analysis; however, the majority (83/89, 93%) were found in both sets.

Methods (pg32): To identify HTALs with a beta-binomial test, we utilized the combined haplotype scripts from WASP. First, we created a CHT input file using the haplotype counts for each loop. Next, we passed these files to `fit_as_coefficients.py` to calculate the binomial overdispersion parameters. Finally, we obtained p-values for each individual separately from `combined_test.py` with the option `-as_only`. These p-values were combined via Fisher's method and both the combined and raw p-values were used for downstream analyses. This analysis resulted in the identification of 89 HTALs, 83 of which were contained in the half normal HTAL set (93%). We repeated the analyses in Figure 6 using the WASP results and observed similar enrichment patterns to the half-normal approach, but found stronger enrichments at imprinted loci (Figure S7).

Reviewer #3 (Remarks to the Author):

The authors have done well to address nearly all the points that I and the other reviewers raised during this revision. In particular, I appreciate the way that they clarified the loop calling and pooling approach that they used, adjusted their language to more accurately represent contact propensity, and tested loop frequency differences between the A and B compartment.

I have a few remaining issues, which are minor, but I felt worth noting.

First, just some very small things:

1) ChIA-PET data does not come from Rao et al, as stated on page 4 of the response to reviewers. The reference does appear to be correctly cited in the text as Tang et al., 2015, but make sure Rao is not cited as the source of this data.

We apologize for this oversight in the response to reviews. We have double checked all citations on the CTCF ChIA-PET in the manuscript and found them to all reference Tang et al.

2) Line 879: "HICCPUS" should be "HICCUPS"

We have fixed this typo.

A bit more substantial issues:

1) I still don't understand the updated explanation of "Normal", "Normal split" and "Abnormal" read pairs in the legend of S1A, and I note that their definition doesn't match the figure (which uses the word "regular", which is not defined). Defining normal as "both ends map to a single restriction enzyme cut site" makes it sound like the ends map to adjacent restriction fragments, both near a single cut site. But I think maybe this means "both ends map uniquely". Then, for "normal split", do you mean that one or both ends are "multiple mapped"- having several potential locations in the genome? And then you're picking the location that gives an expected insert size? Am I correct that you're addressing the issue of unique vs. multi-mapped alignment with these terms? Such as described here: <https://pairtools.readthedocs.io/en/latest/parsing.html> ? Perhaps you're just using your own words for what the Juicer report calls Normal Paired, Chimeric Paired, and Chimeric Ambiguous? If so, why not use those words? The word "chimeric" helps me understand a bit better, but you don't use this word in your legend.

We apologize for the confusion. As the reviewer suggests, these reads are more appropriately called Paired, Chimeric Paired, and Chimeric Ambiguous reads. We have relabeled Figure S1 to use the correct notation. We also modified the figure legend removing definitions of the incorrect wording:

Figure S1: (A) Summary statistics for Hi-C data processing using Juicer pipeline. Green boxes indicate the overall raw read pairs obtained from the Hi-C experiments. Yellow boxes indicate read pairs or contacts retained after each filtering step. Red boxes indicate read pairs removed by filtering. Blue boxes indicate the final retained inter-chromosomal contacts. The percentage values between the boxes represent percentage of read pairs calculated based on the total number of raw read pairs, and the percentage values within the rectangles represent percentages relative to the total number in the box.

2) The authors have done well to switch to the term "contact propensity" rather than "loop

strength” and to clarify what they mean in terms of looping frequency across the population of cells. But, I still think they are equating distance and contacts in a way that is not accurate, or necessary to their conclusions. Indeed, as the authors note, FISH measured distances and Hi-C contact frequency are often inversely proportional, but the strongest relationship is between contact frequency and FISH colocalization (distance less than a threshold). Oddly, as one of the references for the sentence:

“Moreover, fluorescence in situ hybridization studies have shown that spatial distance and contact frequency between looped loci vary concordantly”

the authors cite a paper by Imakaev and Fudenberg. This paper specifically finds “a nontrivial relationship between contact frequency and spatial distance” in which equivalence between the two can be easily broken by dynamic looping contacts—the very situation the authors explore here. Two loops might have the same interaction probability in Hi-C: both occurring in 10% of cells. But if, for one loop, the two partners stay relatively close together when not touching and for the other loop, the two partners are far apart when not touching, the average spatial distance of these two loops would be vastly different.

I hate to belabor this point, because I think the author’s conclusions are valid and interesting solely focusing on what they can measure: contact propensity. They do not have to draw an equivalence to spatial distance in order for their results to be biologically interesting. So, I would recommend that they just stop at the point of explaining what contact propensity is and take out some of their sentences about physical distance between loci.

We agree with the reviewer that the Fudenberg article describes that across paired loci, absolute contact intensities do not reflect absolute differences in spatial proximity. We also agree that the manuscript remains of interest without the context of spatial proximity in the introduction or results. As such, we have removed mentions of spatial proximity in the Introduction and Results, resulting in the following edited Introduction paragraph and removed sentences:

Intro (pg3; removed): Moreover, the extent that regulatory genetic variation acts by affecting the physical distance between looped loci is not yet known.

Intro (pg3): Bulk chromatin conformation assays (e.g. 3C, 4C, and Hi-C) were designed to measure physical contact frequency between two pieces of colocalized (ie looped) DNA in a pool of cells. While a recent single cell Hi-C study found that contacts occur within single cells at loops called from bulk data, there was variability in the contact profiles of looped loci between cells²⁶. Together, this suggests that the contact frequency measured in a pool of cells reflects the proportion of cells in which a contact is occurring, or the probability for the contact to occur (**contact propensity**) across all cells in the sample. Investigating contact frequency as measured by Hi-C, in combination with molecular phenotypes, may reveal if contact propensity between looped loci varies across cell types and haplotypes, and if this is associated with differential regulation of gene expression.

Results (pg12; removed): In the context of spatial proximity, these results suggest that genetic variation could mediate its effects by affecting the spatial proximity between loci colocalized by a chromatin loop.

3) Related to the above, I am uncertain why the authors need to invoke spatial proximity, rather than simply contact frequency, in their discussion of how genetic variation may exert its effects: “Extending our results to spatial proximity, our data suggests that genetic variation at chromatin loops may exert their effects on gene expression or gene regulation by modulating the spatial proximity between the loci of, and thus contact propensity at, chromatin loops.”

I don't think the conclusion has to be extended to overall average spatial proximity to be interesting. I can see a mechanistic explanation for how variation at a specific loop site could affect contact frequency itself: a loss or gain of protein binding due to the variation may make looping interactions more or less favorable/stable. Why do you need to propose that the variant instead works by causing two loci to stay closeby even when not interacting? How would this work? That seems like a property controlled by the chromatin state of the broader region, tethering to nuclear structures, and cohesin-mediated loop extrusion.

We agree with the reviewer that the mechanisms which underly changes in contact propensity is unclear, and that we do not examine these mechanisms in our manuscript. We have therefore removed text that extends contact propensity to spatial proximity. We also agree that there are many likely biological phenomena that could underly changes in contact propensity, including those mentioned by the reviewer (eg protein binding, chromatin state), as well as spatial proximity or regulatory activity. Therefore, in an effort to offer insight into future studies of the mechanism of contact propensity, we have reframed our discussion to discuss potential mechanisms underlying changes in contact propensity and limit our reference to spatial proximity as a potential contributing factor. We hope that these changes satisfy the reviewer and help facilitate future studies examining this phenomenon, while not overstating the findings. The following changes have been made in the discussion:

Discussion (pg12; removed): 3D modeling of the genome from Hi-C²⁶, and fluorescence in situ hybridization experiments^{23,25,53}, have shown a relationship between contact frequency and spatial proximity, which suggests that loci with increased spatial proximity (ie are closer together) come into physical contact more easily, and thus produce a Hi-C contact more readily. Extending our results to spatial proximity, our data suggests that genetic variation at chromatin loops may exert their effects on gene expression or gene regulation by modulating the spatial proximity between the loci of, and thus contact propensity at, chromatin loops. Notably, as we identified a non-directional correlation, contact propensity may either affect, or be affected by, gene expression and/or regulation. In the latter case, genetic variation would mediate effects on spatial proximity through affecting gene expression or regulation.

Discussion (pg12): While the mechanisms underlying changes in contact propensity are currently unknown, there are several reasonable hypotheses. Previous studies showing that the physical 3D structure of the genome can be reconstructed from contact frequency via polymer physics models^{26,48-51} suggest that contact propensity could result from changes in spatial proximity. The fact that CTCF and Pol2 ChIA-PET show similar profiles to Hi-C data⁵ suggest that differences in protein binding near loop loci could also affect contact propensity. Finally, as we found associations between contact propensity and H3K27ac, regulatory chromatin activity

could modulate contact propensity. Future studies examining these mechanisms could provide insights into the biological processes underlying differential contact propensity and gene regulation.

Reviewer #2 (Remarks to the Author):

The authors have substantially improved their manuscript and I have not other concern.

Reviewer #3 (Remarks to the Author):

The authors have thoroughly addressed all of my concerns and I recommend accepting the manuscript.